# Evolution of fast root gravitropism in seed plants

Yuzhou Zhang[1,4], Guanghui Xiao [2,4], Xiaojuan Wang [2,3], Xixi Zhang[1] & Jiří Friml [1]

An important adaptation during colonization of land by plants is gravitropic growth of roots, which enabled roots to reach water and nutrients, and firmly anchor plants in the ground. Here we provide insights into the evolution of an efficient root gravitropic mechanism in the seed plants. Architectural innovation, with gravity perception constrained in the root tips along with a shootward transport route for the phytohormone auxin, appeared only upon the emergence of seed plants. Interspecies complementation and protein domain swapping revealed functional innovations within the PIN family of auxin transporters leading to the evolution of gravitropism-specific PINs. The unique apical/shootward subcellular localization of PIN proteins is the major evolutionary innovation that connected the anatomically separated sites of gravity perception and growth response via the mobile auxin signal. We conclude that the crucial anatomical and functional components emerged hand-in-hand to facilitate the evolution of fast gravitropic response, which is one of the major adaptations of seed plants to dry land.

---

[1] Institute of Science and Technology (IST) Austria, 3400 Klosterneuburg, Austria. [2] College of Life Sciences, Shaanxi Normal University, 710119 Xi'an, China. [3] College of Life Sciences, Northwest University, 710069 Xi'an, China. [4]These authors contributed equally: Yuzhou Zhang and Guanghui Xiao. Correspondence and requests for materials should be addressed to J.F. (email: jiri.friml@ist.ac.at)

Conquest of the land by plants marks one of the most important transition during evolution of life on Earth[1–4]. For plants to thrive in this new environment, number of dramatic developmental adaptations occurred[5]; among them, the evolution of efficient root gravitropic response that allows roots to grow deep into the soil. The early diverging land plants were non-vascular plants without true roots but with the root hair-like organ rhizoids, a structure, which helps plants to attach to the soil surface as an early adaptation to the land environment[6–8]. The fossil evidence indicates that the true roots emerged in the vascular plants[9], and in the flowering plants the root has evolved into an organ to grow downwards along the gravity vector with two main purposes: anchoring in the soil and providing a source of water and nutrients for growth of the above-ground parts of the plants[10].

Root gravitropism of flowering plants is well characterized and comprises three temporally and spatially distinct phases: gravity perception, transmission of the gravitropic signal, and ultimately the growth response itself[11–14]. Unlike in green algae *Chara*, whose root hair-like structure rhizoids utilize the barium sulfate ($BaSO_4$) crystal-containing vacuoles as the gravity-perceiving organelles[15], the gravity perception in flowering plant roots occurs by gravity-induced sedimentation of the dense starch-filled amyloplasts within the specialized columella cells of the root apex. Gravity signal is further transmitted by the intercellular signal auxin with the aid of the auxin importers and exporters from the AUX1/LAX and PIN protein families, respectively[15–20]. Gravity perception leads to the polarization of PIN transporters (PIN3 and PIN7) to the bottom side of columella cells, thus driving the redirection of auxin flow downwards[21–23]. Along the lower root side, mediated by PIN2 protein, auxin is further translocated to the place of auxin response, the elongation zone[13,24–30]. There in the root, unlike in shoots, where auxin promotes growth, auxin rapidly inhibits growth at the lower side and this asymmetry leads to the downward root bending[31–35]. Notably, some findings suggest that, besides the major mechanism of gravity perception by the amyloplast sedimentation in the root cap, there is a secondary, amyloplast-independent site of gravity sensing in the distal elongation zone of flowering plant roots[36].

Despite the profound importance of root gravitropism in plant growth and adaption, most of the related works only focus on the flowering plants, especially the model plant *Arabidopsis thaliana*. The mechanism of root gravitropism has never been systematically compared throughout the plant kingdom and its evolutionary origin remains unknown. Answering this fundamental question would reveal how, during plant evolutionary history, root evolved to be such an efficient device to respond to the Earth gravity.

## Results

**Slow and fast root gravitropism during plant evolution.** To obtain a broad view of the evolutionary origin of root gravitropism, we selected various plant species representing the lineages of mosses, lycophytes, ferns, gymnosperms, and flowering plants, including dicots and monocots, and analyzed their root gravitropic response (Fig. 1). Mosses, including the model *Physcomitrella patens*, have rhizoids but no true roots[37]. After gravistimulation (90° reorientation), the rhizoid showed a much slower gravitropism than the typical roots of flowering plants such as *A. thaliana* (Fig. 1 and Supplementary Fig. 1a). Lycophytes and ferns have a true root, but the model lycophyte *Selaginella moellendorffii* and the model fern *Ceratopteris richardii* showed much slower gravitropism than the roots of the flowering plants *A. thaliana*, *Gossypium arboretum*, or *Oryza*

*sativa* (Fig. 1 and Supplementary Fig. 1b, c). In contrast, the seed plant gymnosperm *Pinus taeda* showed the fast root gravitropism comparable to that of the flowering plants and much faster than that of the lycophyte *S. moellendorffii* and the fern *C. richardii* (Fig. 1). As the growth rates among these diverse plant roots are disparate (Supplementary Fig. 2a), to exclude the effect of the growth rate during the evaluation of root gravitropism, we evaluated the vertical growth index (VGI)[38] on roots with the same root elongation (~2 mm) after the gravistimulation. The results further confirmed the much slower root gravitropism of non-seed plants as compared with that of the seed plants (Supplementary Fig. 2b, c).

This notable difference in the gravitropic efficiency suggest that there are two mechanistically distinct root gravitropic responses: the slow, less efficient gravitropism of basal vascular plant species and the fast root gravitropism, which might have originated in the most recent common ancestor of the gymnosperms and flowering plants after divergence of these seed plants from the basal vascular plant lineages.

**Origin of root apex-exclusive gravity perception.** To determine whether the root architectural innovation may have facilitated the fast root gravitropism in seed plants during evolution, we analyzed the root structures of the representative plant species with a focus on localization of starch-containing amyloplasts (Fig. 2a), which act as the statoliths for the gravity perception in the root of flowering plants[39] (Supplementary Fig. 3a). Lugol's staining for starch granule location of the rhizoids of the moss *P. patent* revealed that they were devoid of amyloplasts (Fig. 2b). In the most primitive living vascular plants, the lycophyte *S. moellendorffii*, amyloplasts have evolved and were found in the root but, interestingly, these starch-filled cells were distributed not within but above the root apex (Supplementary Fig. 4a). In the root of the fern *C. richardii*, the amyloplasts were present both above and within the root apex (Supplementary Fig. 4b). Only in seed plant, the gymnosperm *P. taeda*, the amyploplasts were specifically localized within the root apex (Supplementary Fig. 4c), which is the same as the pattern of amyloplast accumulation in the roots of the flowering plants, the dicots *A. thaliana* and *G. arboretum*, and the monocot *O. sativa* (Supplementary Fig. 4d–f). These results suggest that the amyloplast localization specifically confined to the root apex might have originated in the common ancestor of seed plants only after their divergence from the fern lineage.

To further confirm these results, we performed the modified pseudo-Schiff-propidium iodide staining (mPS-PI) to observe detailed root structure and starch granule localization in these representative plant species. In the lycophyte *S. moellendorffii*, the starch granules (amyloplasts) mainly localized at the two lateral sides of the root above the apex and surrounded by the epidermal cells, but they were absent in both the root apex and the vascular bundle located in the middle of the root (Fig. 2c). In the root of the fern *C. richardii*, the localization of starch granules above the root apex was similar to that observed in *S. moellendorffii*, but they were also present in the root apex below the apical cell, a single large pyramidal and quiescent center (QC)-like cell (yellow arrow in Fig. 2d). Correlating with the observation of fast root gravitropism (Fig. 1a), the starch granules in the gymnosperm *P. taeda* specifically accumulated within the root apex below the QC (Fig. 2e), which is similar to the localization pattern observed in the flowering plants *A. thaliana*, *G. arboretum*, and *O. sativa* (Fig. 2f–h).

In addition, we examined whether the amyloplasts in these basal vascular plant roots served as the gravity-perceiving statoliths of the flowering plants. Notably, the amyloplast sedimentation analysis revealed that in contrast with the

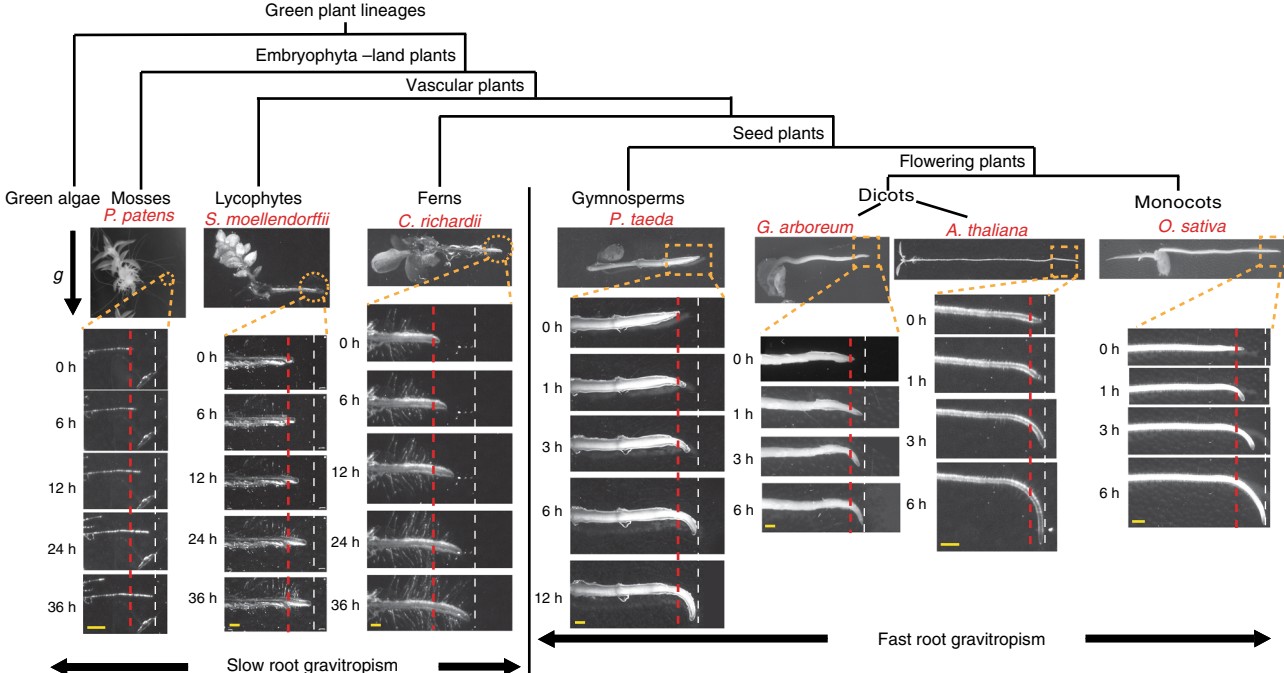

**Fig. 1** Slow and fast mechanisms of root gravitropism during plant evolution. Slow gravitropic bending of the rhizoids of the moss *P. patens*, roots of the lycophyte *S. moellendorffii*, and the fern *C. richardii* after a 90° reorientation of the seedlings. Much faster response of the gymnosperm *P. taeda*, the dicots *G. arboreum* and *A. thaliana*, and the monocot *O. sativa* after gravistimulation. Scale bars, 1 mm

amyloplasts in the root cap of *A. thaliana*, which were mainly located at the basal ends of the cells and showed fast sedimentation after the 180° reorientation, the amyloplasts in the roots of the fern *C. richardii* and lycophyte *S. moellendorffii* showed a random localization in the root cells and failed to sediment after the 180° reorientation (Supplementary Fig. 5a–f). These results strongly indicates that, unlike in flowering plant roots, the gravity-sensing machinery with the amyloplast sedimentation along the gravity vector did not evolve in roots of these basal vascular plants.

All the results above show that the root architectural innovation, in particular root apex-specific amyloplast localization spatially separated from the elongation zone, coincides with the advancement of the fast root gravitropism in seed plants. It suggests that this particular arrangement of gravity perception and growth control has been selected as a strategy for efficient root gravitropism during plant evolution.

**Fast root gravitropism-specific PIN2 of *Arabidopsis*.** In *Arabidopsis*, the directional auxin flow from the apex to the elongation zone is driven by PIN2 auxin transporter that is localized at the shootward sides of root epidermal cells[20,29,30]. PIN2 plays a pivotal role in fast root gravitropism of flowering plant *Arabidopsis*, as disruption of PIN2 blocks the gravity-induced asymmetric auxin redistribution and result in the defective root gravitropism[28] (Fig. 3a and Supplementary Fig. 3b, c).

There are eight *PIN* genes in *A. thaliana* that can be divided into three lineages based on their lengths of hydrophilic loop (HL) and subcellular localizations[40–42]: the canonical, plasma membrane (PM)-localized PINs (PIN1, PIN2, PIN3, PIN4, and PIN7), the endoplasmic reticulum (ER)-localized PIN5 and PIN8, and PIN6 with dual PM and ER localization[40–42]. To determine which of the PINs can mediate the fast root gravitropism, we used the *Arabidopsis PIN2* promoter to drive the expression of the seven *PINs* in a loss-of-function *pin2* mutant (Fig. 3a–d). The non-canonical *PIN6* and *PIN5* were not able to rescue the *pin2*

mutant (Fig. 3c, d), and also of the canonical *PINs* only *PIN2* was able to complement the defective root gravitropism phenotype of *pin2* (Fig. 3b). These results were confirmed by quantification of root gravitropism using the VGI (Supplementary Fig. 6a, b), confirming that only PIN2 can mediate fast root gravitropism in *Arabidopsis*.

**Evolution of PIN2 functionality in fast root gravitropism.** Next, we wanted to know when this PIN2-specific function arose during plant evolution. First, to obtain a broad view of the evolution of PIN2 during the green plant diversification, we used the full-length protein sequence of *Arabidopsis* PIN2 as a query in searches against the available databases for 14 species representing the green algae, the most primitive living land plant marchantiophyta (liverworts), mosses, lycophytes, ferns, gymnosperms, and flowering plants (Supplementary Fig. 7). Then we aligned these PIN protein sequences and constructed a phylogenetic tree (Supplementary Fig. 8). According to the comprehensive PIN phylogeny by Bennett et al.[42], the PIN2 proteins were only present in the flowering plants, which is congruent with our phylogenetic tree. However, it leaves open whether there are PIN proteins in gymnosperms that are functionally similar to the flowering plant PIN2 in root gravitropism.

So to test when the PIN2-specific functionality in root gravitropism has evolved, we performed interspecies genetic complementation experiments with *PIN* genes from various representative plant lineages expressed in *Arabidopsis pin2* mutant under the control of the *Arabidopsis PIN2* promoter. The evolutionary most primitive *PIN* gene known to date from the basal Streptophyte green alga *Klebsormidium flaccidum* (*KfPIN*) was unable to rescue the defects in root gravitropism in the *pin2* mutant (Fig. 3e), although it is a functional auxin transporter (Skokan et al., submitted). Similarly, the single canonical *PIN* (*MpPINZ*) found in the *Marchantia polymorpha*[42,43], a probable representative of the earliest diverging land plants[6], also failed to complement the defective *pin2*

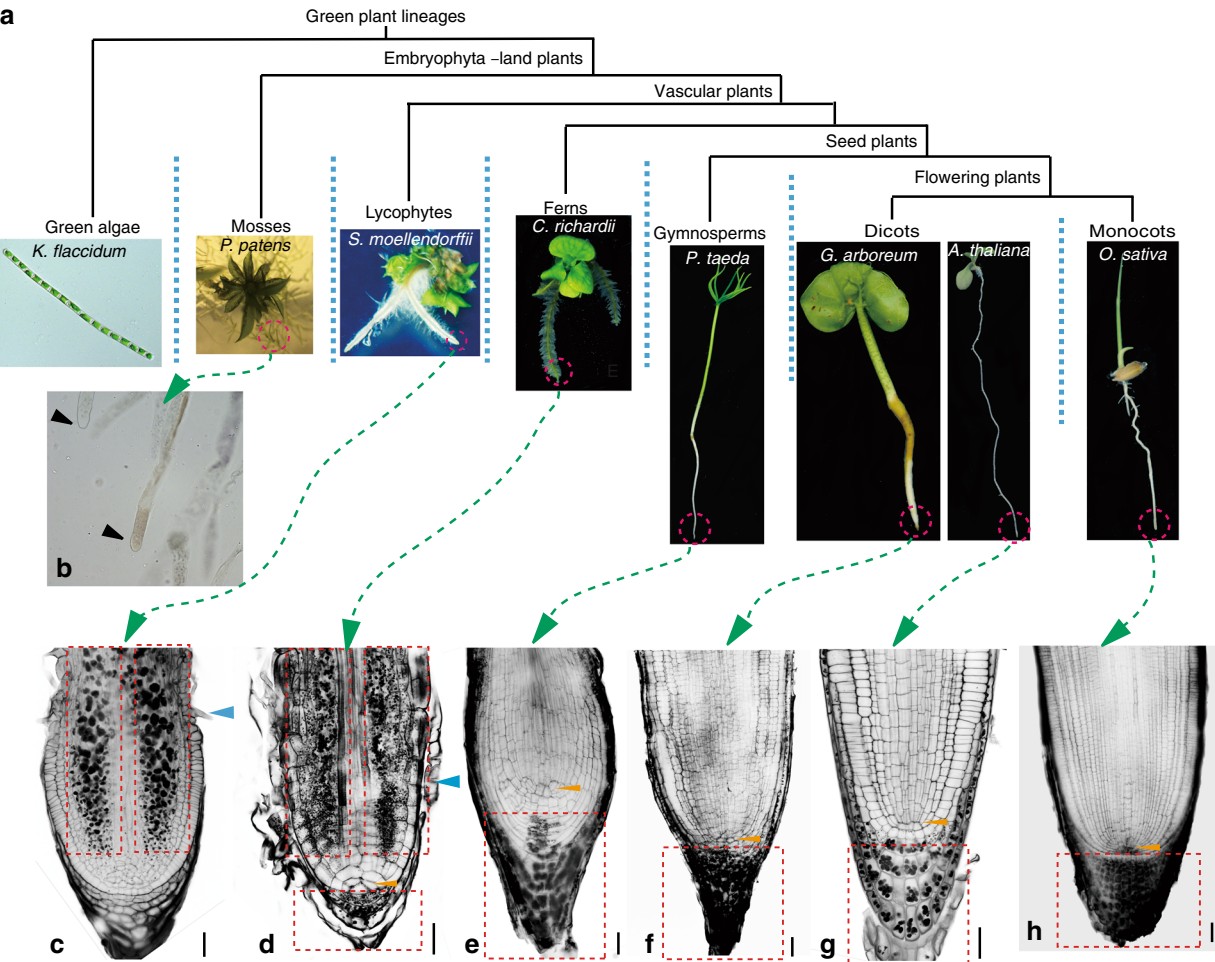

**Fig. 2** Exclusive root apex-specific amyloplast localization in seed plants. **a** Living representative species from different plant lineages included in the analysis (from left to right): *K. flaccidum* (green alga), *P. patens* (moss), *S. moellendorffii* (lycophyte), *C. richardii* (fern), *P. taeda* (gymnosperm), *G. arboreum*, and *A. thaliana* (dicots), and *O. sativa* (monocot). **b** Lugol's staining of the rhizoid (*P. patens*). **c–h** mPS-PI staining of the root tips from *S. moellendorffii* (**c**), *C. richardii* (**d**), *P. taeda* (**e**), *G. arboreum* (**f**), *A. thaliana* (**g**), and *O. sativa* (**h**). The blue arrows indicate root hair initiation. The yellow arrows indicate the apical cell (QC-like cell) in the fern *C. richardii* and the QC in seed plants. The dashed red rectangles indicate the zone with amyloplasts. Scale bars, 20 μm

root gravitropism (Fig. 3f). Representative canonical PINs from the non-vascular plant, the moss *P. patens* (i.e., *PpPINA* and *PpPINB* from clade 6), the basal vascular plants, the lycophyte *S. moellendorffii* (i.e., *SmPINR* and *SmPINU* from clade 6), and the fern *C. richardii* (i.e., *CrPINJ* and *CrPINN* from clade 7), all failed to replace the fast root gravitropism function of *Arabidopsis AtPIN2* (Fig. 3g–i and Supplementary Fig. 9). In the basal seed plant gymnosperm *P. taeda* (Pt), we identified five *PIN* genes, distributed in the five clades of the PIN phylogeny[42], but domain prediction clearly indicated that the PtPINF protein is not complete. Therefore, we cloned the four *PIN* genes from the other four clades to perform the interspecies complementation experiments. In contrast to other *PIN* genes from the *P. taeda*, only two, *PtPINH* and *PtPING*, were able to rescue the defective root gravitropism phenotype of the *Arabidopsis pin2* mutant (Fig. 3j). The quantitative PCR analysis revealed that the two *PIN* genes of *P. taeda*, *PtPING* and *PtPINH*, were strongly expressed in the root tip as compared with shoot and the other part of the root (Supplementary Fig. 10a–c), thus resembling the expression pattern of *PIN2* in the flowering plant root. The auxin transport assay with [3]H-labeled indoleacetic acid ([3]H-IAA) showed efficient shootward auxin transport from the root tip of *P. taeda* that was sensitive to the N-1-naphthylphthalamic acid (NPA), an established inhibitor of auxin transport[44] (Supplementary

Fig. 10d, e). The shootward auxin transport efficiency in the root of fern *C. richardii* is much lower than that in the *P. taeda* and largely NPA-insensitive (Supplementary Fig. 10e). These results suggest that the efficient shootward auxin transport along with the required functional PIN proteins for fast root gravitropism have originated in the seed plants after the divergence from the basal vascular plant lineages. Notably, in flowering plants, such as *Arabidopsis* and *O. sativa*, there is only one *PIN2* gene, whereas there are two *PIN* genes in gymnosperm *P. taeda* and *P. abies* with the functional equivalent to the *Arabidopsis PIN2*, suggesting that a duplication event of the *PING/H* progenitor occurred during the evolution of gymnosperms.

Heterologous expression of the *PIN2* gene from the flowering plant *G. arboretum* (*GaPIN2*) in *Arabidopsis* successfully complemented the *Arabidopsis pin2* mutant phenotype, indicating that this protein is functionally equivalent to *Arabidopsis AtPIN2* (Fig. 3k–m). A recent report showed that the monocot rice *PIN2* gene (*OsPIN2*) also could rescue the *Arabidopsis pin2* mutant phenotype[45]. These successful interspecies complementation experiments imply that the unique PING/H and PIN2 with fast gravitropic function appeared and evolved in gymnosperm and flowering plant lineages since the separation of the seed plants from the vascular plants.

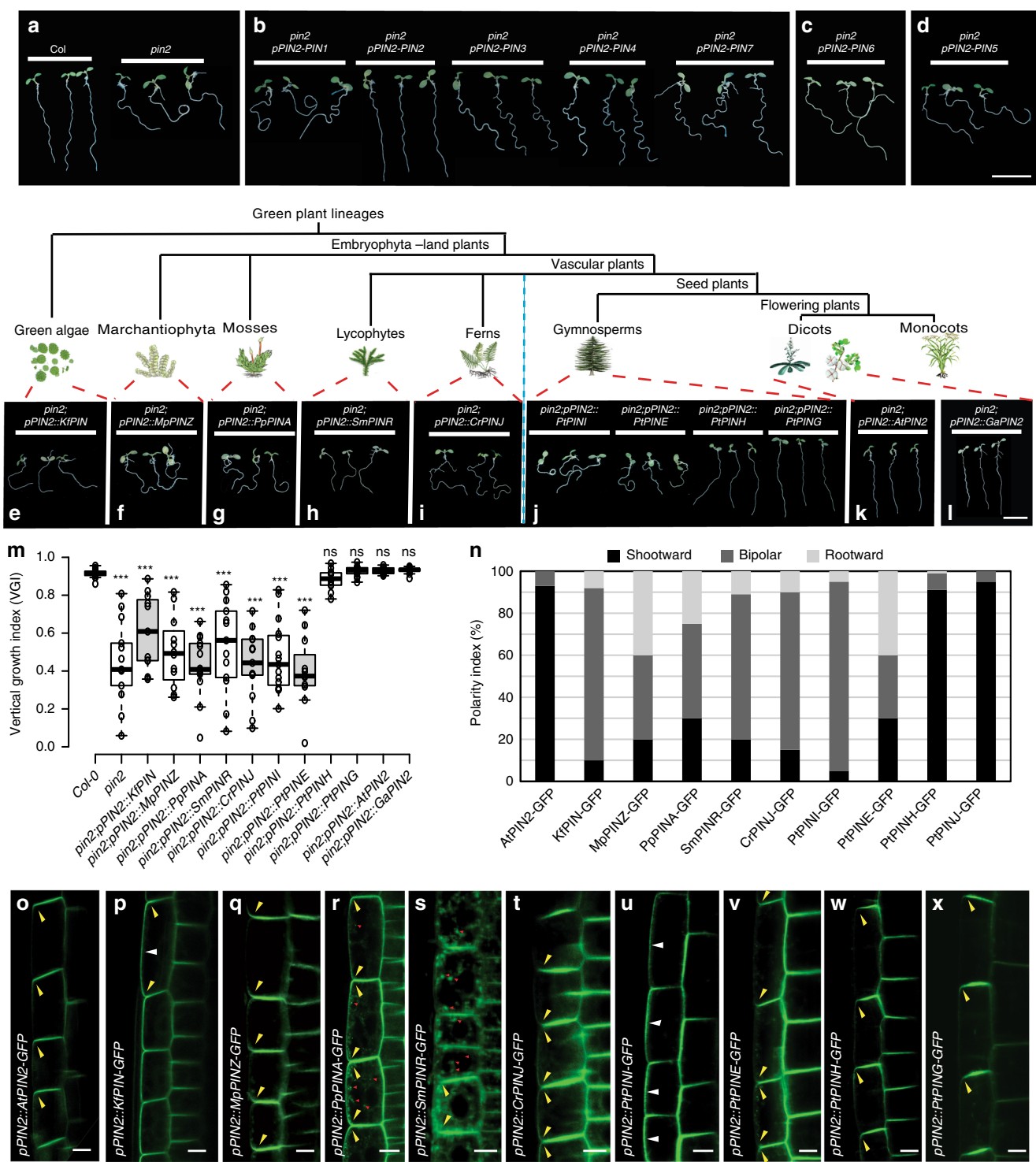

**Origin of apical PIN localization for shootward auxin flow.** We hypothesized that the shootward subcellular localization of the PIN proteins was the innovation leading to fast gravitropism. To confirm this, we analyzed the root epidermal cell localization of a series of PIN-GFP fusion proteins driven by the *Arabidopsis* native *PIN2* promoter. In contrast to the *Arabidopsis* AtPIN2-GFP fusion protein, which is predominantly localized at the shootward side of the epidermal cells (Fig. 3n, o), the green alga KfPIN-GFP showed non-polar and also lateral localization in these cells (Fig. 3n, p). The marchantia MpPINZ-GFP, the moss PpPINA-GFP and PpPINB-GFP, and the lycophyte SmPINR-

GFP fusion proteins showed non-polar localization at the PM of the *Arabidopsis* root epidermal cells and occasionally aggregated granules of these PIN proteins in the cytoplasm were also observed (Fig. 3q–s and Supplementary Fig. 11). Interestingly, the PpPINA and PpPINB proteins showed obvious polar cellular localization in the moss *P. patens* rhizoid[43] (Supplementary Fig. 12) but they did not acquire the specific ability to localize at the shootward side of cells. The fern CrPINJ-GFP fusion protein showed the predominately bipolar localization in *Arabidopsis* root epidermal cells (Fig. 3n, t). The gymnosperm PtPINI-GFP proteins showed bipolar and strong lateral localization in the

**Fig. 3 Origin of fast root gravitropism-specific PIN2 functions in seed plants. a–d** In contrast to the wild type (Col), the *pin2* mutant showed severe defects in root gravitropism (**a**). Genetic complementation experiments showing that of the *A. thaliana* non-canonical (*PIN5* and *PIN6*) and canonical (*PIN1*, *PIN2*, *PIN3*, *PIN4*, and *PIN7*) PINs, only PIN2 rescues the defective *pin2* gravitropism (**b–d**). Scale bars, 1 cm. **e–m** Interspecies complementation experiments with orthologous *PIN2* genes from green alga (*KfPIN*) (**e**), marchantiophyte (*MpPINZ*) (**f**), moss (*PpPINA*) (**g**), lycophyte (*SmPINR*) (**h**), fern (*CrPINJ*) (**i**), gymnosperm (*PtPINI*, *PtPINE*, *PtPINH*, and *PtPING*) (**j**), *Arabidopsis* (*AtPIN2*) (**k**), and *G. arboreum* (*GaPIN2*) (**l**). Only the gymnosperm genes encoding PtPINH and PtPING (Supplementary Fig. 8), and the flowering plant genes encoding AtPIN2 and GaPIN2 from the PIN2 clade were able to rescue the *pin2* defects in root gravitropism (**k–l**). Scale bars, 1 cm. **m** Quantification of VGI for the plants in **e–l** ($n \geq 10$ roots). Center lines show the medians; box limits indicate the 25th and 75th percentiles as determined by R software; whiskers extend 1.5 times the interquartile range from the 25th and 75th percentiles, outliers are represented by dots. Student's *t*-test, \*\*\**P* < 0.001 and ns denotes *P* > 0.05, compared with the *Col-0*, respectively. **n–x** The subcellular localization of AtPIN2 (**o**), KfPIN (**p**), MpPINZ (**q**), PpPINA (**r**), SmPINR (**s**), CrPINJ (**t**), PtPINE (**u**), PtPINI (**v**), PtPINH (**w**), and PtPING (**x**) in *Arabidopsis* root epidermal cells. Only the gymnosperm genes encoding PtPINH and PtPING (**w, x**), and the flowering plant genes encoding AtPIN2 from the PIN2 clade (**o**) were able to localize to the shootward cell side. The coding sequences were fused with GFP in the central HL and expression was driven under the control of the *Arabidopsis PIN2* promoter. Polarity index of the cellular localization of the PIN-GFP fusion proteins ($n = 150$–$200$ cells from ten roots) (**n**). Scale bars, 10 μm. Source data are provided as a Source Data file

*Arabidopsis* root epidermal cells (Fig. 3u), whereas most of the PtPINE-GFP fusion proteins showed rootward/bipolar localization (Fig. 3v). Only the gymnosperm proteins PtPINH-GFP and PtPING-GFP were predominantly localized at the shootward side of the root epidermal cells (Fig. 3w, x), which correlates with their ability to complement the *Arabidopsis pin2* mutant phenotype (Fig. 3j), and efficient shootward auxin transport in the root tip of *P. taeda* (Supplementary Fig. 10d, e).

Moreover, the amino acid sequence alignments revealed that the key phosphorylation sites of *Arabidopsis* PIN2 (AtPIN2), which were identified in its central HL and critical for PIN2 shootward subcellular localization and its function in root gravitropism[46], have been evolutionarily conserved in other flowering plant PIN2 and gymnosperm PING/H (Supplementary Fig. 13). This is consistent with the shootward localization and conserved function of these PIN2-like proteins in root gravitropism as revealed by the successful interspecies complementation experiments.

Our results indicate that during the evolution of land plants, the specific shootward cellular localization of PIN2 appeared together with the efficient shootward auxin transport and fast gravitropic response was among the crucial innovations, which endowed PIN2 with its specific ability to mediate this process in seed plants.

**Functional innovations of PIN2 during plant evolution.** Our analysis revealed that the shootward localized PIN protein, which mediates fast root gravitropism, evolved only in seed plants. However, it is still unclear what functional innovations at the sequence level were important for the unique PIN2 function in fast root gravitropism.

The intergenic domain swapping experiments combined with interspecies complementation experiments showed that when the central HL of the green alga KfPIN was replaced by the HL of the *Arabidopsis* AtPIN2 (Fig. 4a, upper panel), the hybrid PIN protein (denoted as X1) still failed to complement the *Arabidopsis pin2* mutant root gravitropism phenotype (Fig. 4a, middle panel). Consistent with this, the hybrid PIN (X1-GFP) also showed abnormal cellular localization (shootward/bipolar localization) in *Arabidopsis* root epidermal cells (Fig. 4a, lower panel), suggesting that the transmembrane domains (TMDs) of PIN also contribute to the regulation of the PIN2 polar localization and its function in root gravitropism (Fig. 4h). However, when the central HL of MpPINZ from the primitive living land plant was replaced by the central HL of AtPIN2, this hybrid PIN (denoted as X2) X2-GFP fusion protein was predominantly localized at the shootward side of *Arabidopsis* root epidermal cells and was able to rescue the defective root gravitropism phenotype of the *Arabidopsis pin2* mutant (Fig. 4b). These results, combined with the observation of

the X1-GFP protein property (Fig. 4a), strongly suggests that a functional innovation in the TMDs of the PIN2 predecessor occurred in the common ancestor of land plants after their divergence from the green alga lineage.

Moreover, when we replaced the central HL of the lycophyte SmPINR or the fern CrPINJ with the *Arabidopsis* AtPIN2 central HL (Fig. 4c, upper panel), both hybrid PINs (denoted as X3 and X4, respectively), similar to *marchantiophyta* hybrid X2 also showed shootward localization in *Arabidopsis* epidermal cells and were able to rescue the impaired gravitropism phenotype of *pin2* (Fig. 4c, middle/lower panels). These results suggest that after the first functional innovation of the ancestral PIN protein in the early diverging land plants, the function of the canonical PIN TMDs in root gravitropism has been evolutionarily conserved during the evolution of the vascular plants after they split from the bryophyte lineages.

We postulated that another functional innovation of PIN2 occurred in its central HL in the most recent common ancestor of the seed plants (gymnosperms and flowering plants), because the gymnosperm PtPING-GFP and flowering plant AtPIN2-GFP fusion proteins were able to rescue the *pin2* mutant phenotype, and these proteins predominantly localized at the shootward side of the *Arabidopsis* root epidermal cells (Fig. 4d), indicating that not only functional TMDs but also a functional HL, both contributing to their polar localization and fast root gravitropism, were acquired by these PIN2 proteins. To further test our hypothesis that the functional PIN2 central HL in root gravitropism originated in seed plants after the divergence from the fern lineage, we fused the central HL of PtPING from the gymnosperm *P. taeda* with the TMDs of the fern protein CrPINJ, and investigated its function in root gravitropism (Fig. 4e, upper panel). This hybrid PIN (denoted as X5) was able to complement the defective root gravitropism phenotype of the *Arabidopsis pin2* mutant and also showed the shootward localization in root epidermal cells (Fig. 4e–g, middle/lower panels).

These results further confirmed that the PIN shootward localization was the crucial functional innovation enabling PIN2 to mediate fast root gravitropism in flowering plants. It also shows that this occurred through in two steps during plant evolution: (i) early (at the onset of land plants) functional innovations in the TMD and (ii) later innovations (after the divergence of the seed plants) in the central HL.

## Discussion

Our systematic comparison of the root bending dynamics in different species revealed at least two distinct modes of root gravitropic response in the plant kingdom: (i) slow, rudimentary response of early diverging vascular plant lineages (lycophytes and ferns) and (ii) much more effective, faster gravitropic

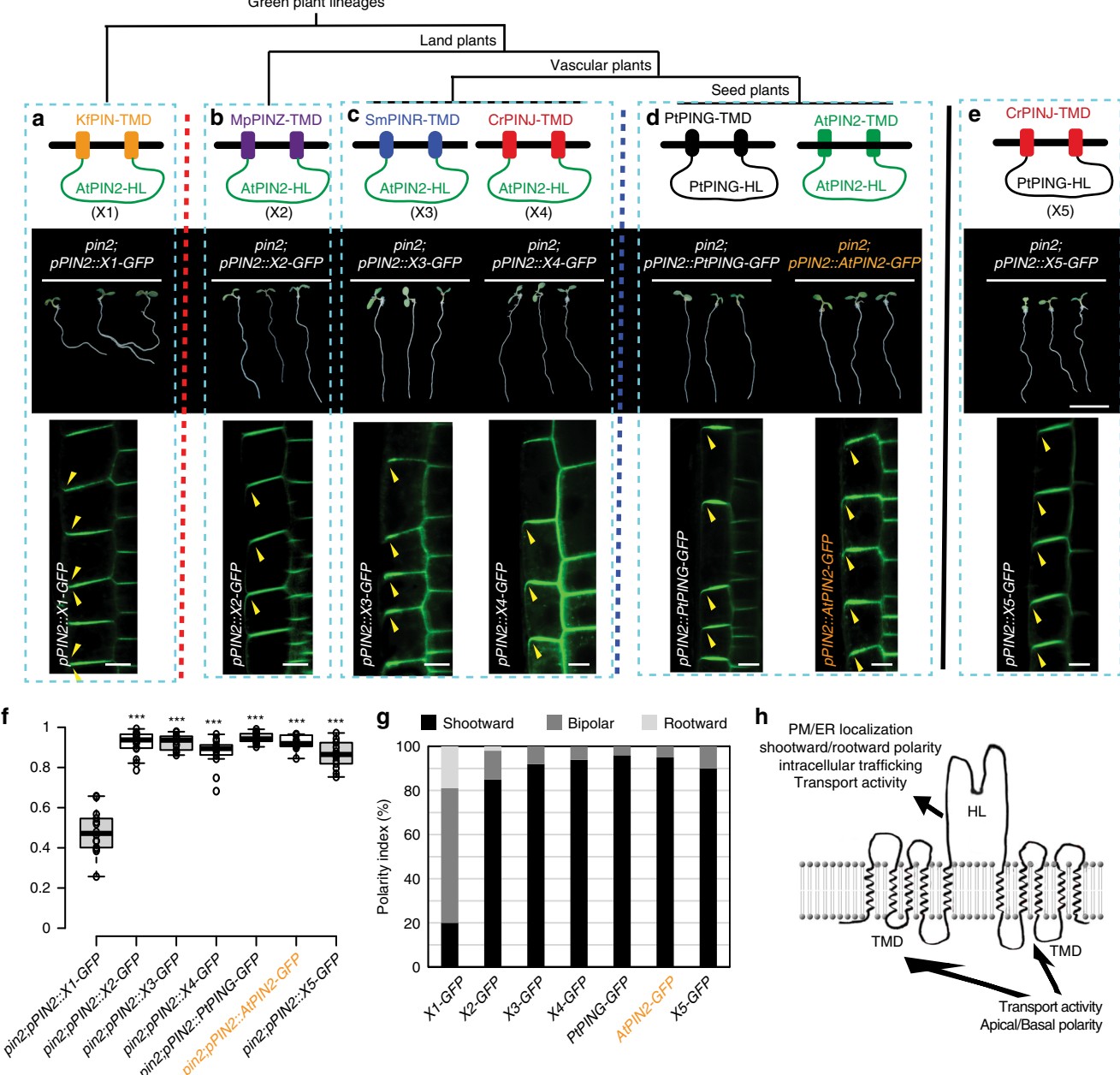

**Fig. 4** Functional innovations of PIN2 during plant evolution. **a–e** Hybrid PIN (X1) with the central hydrophilic loop (HL) of green alga KfPIN substituted by the *Arabidopsis* AtPIN2 HL fused with GFP (HL-GFP) failed to rescue the *pin2* mutant (**a**, middle panel). Hybrid PIN proteins (X2-X4) with the central HL of marchantiophyte MpPINZ, lycophyte SmPINR, and fern CrPINJ replaced by the *Arabidopsis* AtPIN2 HL-GFP were able to rescue the defective root gravitropism phenotype of the *Arabidopsis pin2* mutant (**b**, **c**, middle panels). The PtPING and AtPIN2 proteins with GFP fused with the central HL can complement the *pin2* mutant phenotype (**d**, middle panel). Scale bars, 1 cm. The hybrid PIN protein (X5) with the central HL of CrPINJ replaced with the gymnosperm PtPING HL fused with GFP successfully complemented the *Arabidopsis pin2* mutant phenotype (**e**, middle panel). The cellular localization of hybrid PIN proteins (X1–X5) and the PIN2-GFP and PtPING-GFP fusion proteins in *Arabidopsis* root epidermal cells with expression driven by the *AtPIN2* promoter (**a–e**, lower panels). All and only PIN2 chimeric proteins that were able to localize to the shootward cell side also rescued the *pin2* gravitropism. Scale bars, 10 μm. The red and blue dashed lines indicate the separate two functional innovations at different plant evolution stages. **f** Quantification of vertical growth index (VGI) for the transgenic lines in **a–e** (*n* ≥ 10 roots). Center lines show the medians; box limits indicate the 25th and 75th percentiles as determined by R software; whiskers extend 1.5 times the interquartile range from the 25th and 75th percentiles, outliers are represented by dots. Student's *t*-test, ***$P < 0.001$, compared with the *pin2;pPIN2::X1-GFP*, respectively. **g** Polarity index of the cellular localization of the PINs in epidermal cells in (**a–e**, lower panels) (*n* = 150 to 200 cells from ten roots). **h** Schematic showing the contributions of the TMDs and the HL domain to PIN2 functions. Source data are provided as a Source Data file

bending of roots from seed plants (gymnosperm and flowering plants). This difference in the gravitropic functionality may be related to the independent evolutionary origin of roots in these plant groups[47] and also correlates with the anatomical innovations of root architecture, in particular the presence of gravity-sensing statoliths exclusively confined to the root apex, which we detected only in seed plants. As a consequence, the apex-specific place of gravity perception in the root cap has

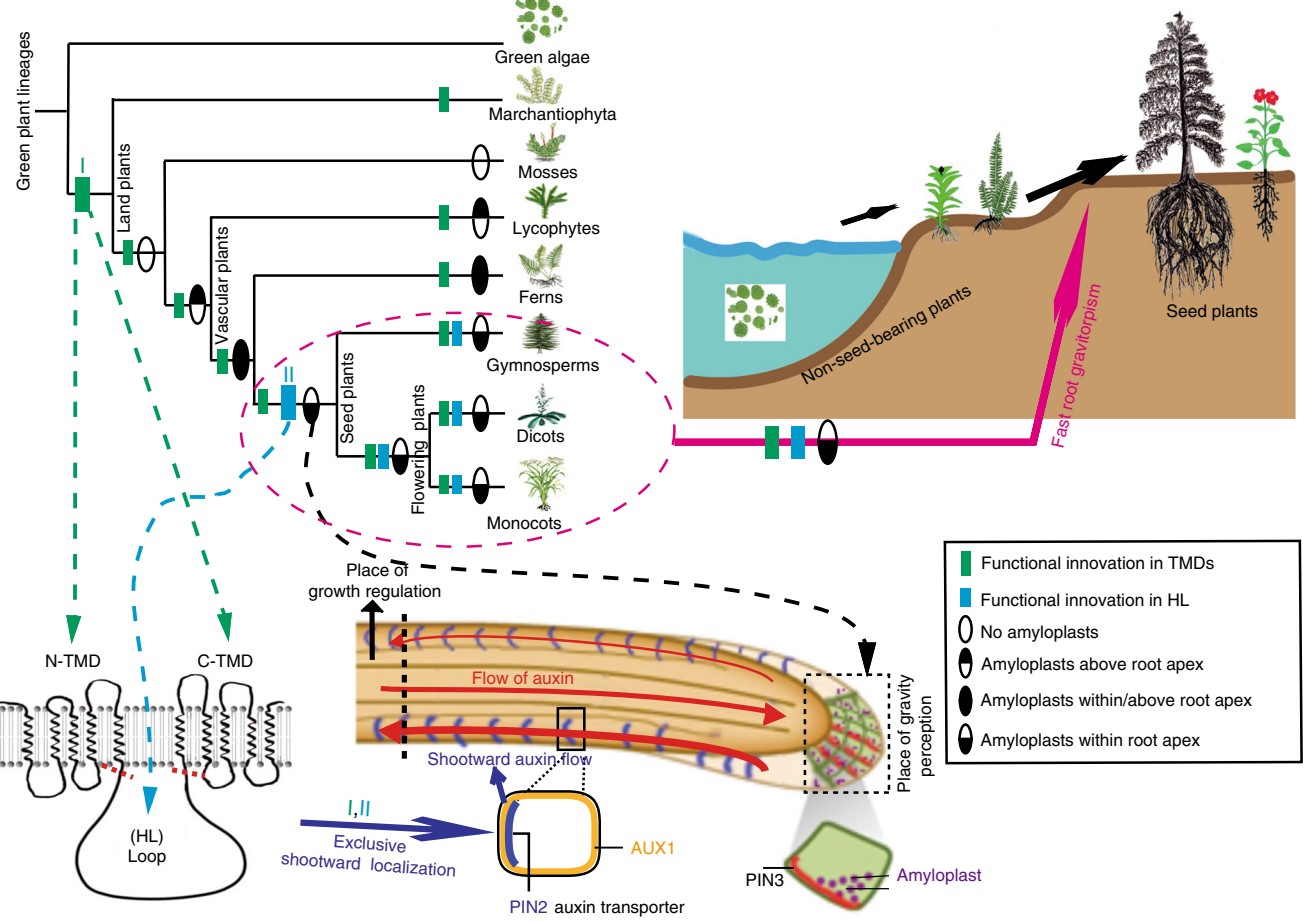

**Fig. 5** Origin of fast root gravitropism facilitated seed plant adaption to the dry land. A schematic diagram representing an evolution of fast root gravitropism that can be detected only in seed plants. It depicts (i) the root anatomical innovation with evolution of the apex-specific gravity perception that is spatially separated from the elongation zone where differential growth occurs and (ii) the functional innovations of the auxin transporter PIN2 that occurred in two disparate plant evolution stages and in different parts of the PIN2 protein sequence ultimately leading to the shootward subcellular localization of PIN2. This novel PIN2 property endowed roots with the ability to transport auxin shootwards enabling the auxin-based signaling from the place of gravity perception to the zone of growth regulation. The fast root gravitropism has evolved as a result of these concomitant anatomical and functional innovations exclusively in seed plants as one of the important adaptions to dry land

evolved to be separated from the place of the growth response necessitating a new signaling mechanism between these tissues (Fig. 5). This has been enabled by the evolution of a new type of PIN auxin transporter, which might be driven by the positive natural selection (Supplementary Fig. 14) and was able to localize to the shootward side of root epidermal cells. This functional innovations occurred first at the onset of the land plants in the PIN2 TMDs and later with advancement of the seed plants in its central HL. These sequence changes resulted in the exclusively shootward, subcellular localization of PIN2 to establish a new, shootward auxin transport flow connecting the place of gravity perception in the root cap and growth regulation in the elongation zone (Fig. 5).

Our genetic complementation experiments suggested that the PING/H in gymnosperm and the PIN2 in flowering plants evolved the similar biological property in mediating the fast root gravitropism, but it is still unresolved whether this is resulted from the convergence evolution of PIN2 and PING/PINH or because they originated from a shared descent. The protein-level phylogenetic analysis with PIN sequences from hundreds of representative plant species showed that the PING/PINH and PIN2 can be grouped in the same clade[42], supporting that gymnosperm PING/PINH are the co-orthologs of flowering plant PIN2 and they may have originated in the most recent common

ancestor of the seed plants. However, the nucleotide-level phylogenetic analysis of the PIN family and modular structure analysis of the HL suggested that the function of PINH/PING and PIN2 in root gravitropism evolved by convergence in gymnosperms and flowering plants[42].

In the flowering plant *Arabidopsis*, besides the AtPIN2, the AtPIN3, and AtPIN7 from the PIN3 clade (Supplementary Fig. 8), which are localized at the bottom side of the gravity-sensing statocytes, are also involved in the root gravitropism. Following a gravitropic stimulus, AtPIN3 and AtPIN7 rapidly relocalize laterally within the first few minutes to facilitate the asymmetrical auxin redistribution between the upper and lower parts of the root[21,22] (Fig. 5). Interestingly, the recent protein-level phylogenetic analysis revealed that the seed plant gymnosperm PINE can be grouped in the same clade with the PIN3[42]. Moreover, according to the phylogenetic tree, the PINE/PIN3 clade is clearly absent in the non-seed plant species. These results are congruent with our observation that the fast root gravitropism has evolved in the seed plants rather than in the non-seed plants, which strongly suggests that also this PIN clade (PINE/PIN3) evolved to facilitate the fast root gravitropism of the seed plants after their divergence from the fern lineage. Notably, in some of the monocots (e.g., *O. sativa* and *Z. mays*), the PIN3 clade is missing as well (Supplementary Fig. 8). Given that the number of

PIN family members in monocot dramatically expanded (Supplementary Fig. 7b), some of the other monocot PIN clades presumably replaced the PIN3 function in root gravitropism during the evolution.

The seed plants, which may have evolved in the late Devonian around 370 million years ago and represents a remarkable life-history transition for photosynthetic organism, underwent dramatic evolutionary radiations and became the dominant group of vascular plants in most habitats. Compared with their predecessors, the seed plants evolved numerous characteristics to facilitate their adaption, such as seed organs, which allowed them to break their dependence on water for reproduction and embryo development[5,48–50]. Our work demonstrates how root anatomical innovation, combined with the evolution of *PIN* auxin transporters, led to the evolution of the seed plant root to become a delicate and efficient organ to mediate fast gravitropism, which might have facilitated their adaption to the new environment along with numerous other evolved traits (e.g., hydrotropism and other growth behaviors).

## Methods

**Search for PIN family members**. The PIN coding sequences (CDS) in the following plants were identified by using the *A. thaliana* PIN2 protein sequence as query in BLAST searches against Phytozome (https://phytozome.jgi.doe.gov/pz/portal.html#!search?show=BLAST): *M. polymorpha*, *P. patens*, *S. moellendorffii*, *A. thaliana*, *O. sativa*, and *Zea mays*. The CDS sequences of *MvPIN* in *Mesostigma viride* and *KfPIN* in *K. flaccidum* (UTEX strain #321; GenBank number: KJ466099) were obtained from the unpublished transcriptome database provided by E. D. Cooper and C. F. Delwiche. The complementary DNA sequence of *C. richardii* PIN was obtained from the transcriptome sequences of *C. richardii* (Jody Banks unpublished data). The PIN sequences of *Cystopteris fragilis* was identified from the 1KP project database (https://db.cngb.org/onekp/). The PIN sequences of *P. abies* and *P. taeda* were identified from the Spruce Genome Project database (http://congenie.org/start). The PIN CDS of *Amborella trichopoda* was identified from the *Amborella* database (http://amborella.huck.psu.edu/wwwblast). The accession numbers/IDs of the identified *PIN* genes are given in Supplementary Table 1. The accession numbers/IDs of the PIN proteins in *G. arboretum* can be found in Zhang et al.[51].

**Evolutionary analysis**. *PIN* genes were translated into protein sequences and subsequently aligned with ClustalX[52]. Neighbor-joining (NJ) and maximum-parsimony (MP) phylogenetic analyses were conducted with MEGA 7[53]. Maximum-likelihood (ML) phylogenetic analysis was conducted with PhyML v3.0[54]. NJ analysis was performed using the protein Poisson distances and the pairwise deletion of gap sites. The default parameters were used for MP analysis. The best-fitting substitution model for the ML analysis was selected with the jModelTest2 program[55]. For each of three phylogenetic analyses, 1000 bootstrap replicates were performed to evaluate the reliability of the phylogenetic trees.

**Plant materials and growth conditions**. Protonemal tissue of the moss *P. patens* was subcultured several times for a minimum of 7 days on cellophane-covered plates with BCD medium containing 5 mM ammonium tartrate and 0.8% agar. Growth conditions were as follows: 24 °C in a long-day light regime, light intensity 55 μmol m$^{-2}$ s$^{-1}$. *K. flaccidum* plants were grown on solid or in liquid M-medium[56] with no sucrose added. The growth conditions were the same as those for *P. patens*. Unless stated otherwise, other plant species were grown vertically in Petri dishes on 0.5× Murashige and Skoog (MS) medium (pH 5.9) containing 1% sucrose and 0.8% agar, at 18 °C under a long-day light regime (light intensity: 250 μmol m$^{-2}$ s$^{-1}$). *A. thaliana*, *P. taeda*, and *S. moellendorffii* were grown at 22 °C, whereas *C. richardii*, *G. arboreum*, and *O. sativa* were grown at 30 °C. The *Arabidopsis* loss-of-function mutant *pin2* and the starchless mutant *pgm-1* were previously described[39,57]. The VGI of *Arabidopsis* root was measured as previously described[38]. For microscopic analyses of gravitropism, seedlings grown in Petri dishes containing 0.5× MS medium were gravi-stimulated by rotating the stage 90° for the specified amount of time before imaging. Bending angles were measured by ImageJ for more than 60 seedlings per genotype (NIH; http://rsb.info.nih.gov/ij).

**Shootward auxin transport assay**. The seedlings were placed on new 0.5× MS plates. Three seedlings for each plant species or treatment with three replicates. Fifteen microliters of $^3$H-IAA was added into 10 mL of 0.5× MS medium with 1.25% agar to make a final 5 μM $^3$H-IAA and then incubated at 65 °C. Five miroliters of $^3$H-IAA droplet was placed on the root apex for 6 h in the dark. N-1-Naphthylphthalamic acid (NPA; 10 μM) was used as a control. There independent experiments were carried out with a similar significant results.

**Vector construction and complementation analysis**. To generate plasmids for genetic complementation analysis, *PIN* CDS from different plant species and 1.4 kb *PIN2* promoter were separately cloned into the Gateway entry vector pDONR221 and pPONRP4P1r vector by BP reaction, and then they were fused and cloned into Gateway destination vector pB7m24GW.3 by LR reaction. To construct the PIN-GFP fusion proteins, GFP was fused in-frame to the central HL of various PIN open reading frames by performing overlapping PCR and the PCR products were then cloned into the Gateway vector pB7m24GW.3 containing the *Arabidopsis PIN2* promoter as described above. The primers used to generate these constructs are detailed in Supplementary Data 1. Transgenic *Arabidopsis* plants were generated using the floral dip method and selected on solid, half-strength MS medium containing 15 mg/mL of Basta (Glufosinate).

**Starch staining**. *Arabidopsis* roots (7 days old) were dipped in Lugol's staining solution (Sigma-Aldrich) for 5 min, washed with distilled water, and then observed under a differential interference contrast microscope (Leica DMRE). The starch granules and cell walls in *Arabidopsis* root tips (7 days old) were stained using the mPS-PI method and imaged with a confocal microscope as previously described[58]. In brief, whole seedlings were fixed in 50% methanol/10% acetic acid at 4 °C for up to 24 h. The tissue was rinsed briefly with ddH$_2$O and incubated in 1% periodic acid at room temperature for 40 min. The tissue was then rinsed twice with ddH$_2$O and incubated in Schiff reagent with PI (100 mM sodium metabisulphite, 0.15 N HCl, and 100 mg/mL PI) for 2 h until the plants were visibly stained. More than three samples were transferred onto microscope slides and covered with chloral hydrate solution (4 g chloral hydrate, 1 mL glycerol, and 2 mL water). The slides were kept overnight at room temperature, after which excess chloral hydrate was removed. The seedlings were mounted in Hoyer's solution (30 g gum arabic, 200 g chloral hydrate, 20 g glycerol, and 50 mL water). The slides were left undisturbed for at least 3 days before observation (excitation 488 nm and emission 520–720 nm).

**Reporting summary**. Further information on research design is available in the Nature Research Reporting Summary linked to this article.

## Data availability

The data that support the findings of this study are available from the corresponding author upon reasonable request. The source data underlying graphs and gels are provided as a Source Data file.

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

## Acknowledgements

We thank J. Banks for providing *C. richardii* spores and PIN sequences, and valuable comments on the manuscript; K. Wang for seeds of *Gossypium arboreum* and *O. sativa*. We also thank E. Medvecká for the KfPIN-GFP fusion protein construct; E. D. Cooper and C. F. Delwiche for the MvPIN sequence and the KfPIN. The research leading to these results has received funding from the European Research Council under the European Union's Horizon 2020 research and innovation Programme (ERC grant agreement number 742985), Austrian Science Fund (FWF, grant number I 3630-B25) and IST Fellow program.

## Author contributions

J.F. and Y.Z. conceived the research and designed the experiments. Y.Z. and X.Z. performed the experiments. G.X. and X.W. performed the bioinformatics analysis. J.F. and Y.Z. wrote the manuscript, and G.X. and X.W. edited the manuscripts. All authors contributed to the manuscript and discussed the results extensively.
