## [Peer Review File · Nature Communications]

Reviewers' comments:

Reviewer #1 (Remarks to the Author):

The manuscript by Yuzhou Zhang and collaborators (NCOMMS-19-04573-T: Evolution of Efficient Root Gravitropism During Conquest of Land by Plants) reports on investigations aimed at better understanding the evolution of the machinery associated with efficient, fast root gravitropism in flowering plants. The authors note that the specific accumulation of starch grains in the root apex, physically separated from the graviresponding elongation zone, appeared coincidentally with fast root gravitropism in land plants. They point out that the need to transport auxin from the site of sensing to the site of curvature response (the elongation zone) was fulfilled by appearance of a shootward-localized efflux carrier, PIN2. They develop a phylogenetic analysis of plant PIN proteins, which shows the existence of clearly separated clades containing PIN2 proteins in the flowering plants (named clade 4) and another one containing gymnosperm PIN2-like proteins (clade 3). Using transformation-rescue experiments of *Arabidopsis* pin2 knockout plants, they demonstrate that angiosperm PIN2 and gymnosperm PIN2-like proteins, are capable of fully rescuing the gravitropic defect of the mutant, whereas other *Arabidopsis* paralogous PINs belonging to different clades (PIN1, for instance), or PIN proteins found in non-seed plants, cannot rescue the *Arabidopsis* mutant. Furthermore, replacing the central hydrophilic loop of these heterologous proteins with that from *Arabidopsis* PIN2 led to rescue as long as the initial protein derived from a land plant (the fusion protein carrying the green algae transmembrane domains was not able to rescue the mutant phenotype in *Arabidopsis*). Similarly, replacing the central hydrophilic loop from these proteins by those from gymnosperms, allowed rescue of the gravitropic response in pin2 mutant roots. In all these transformation-rescue experiments, gravitropism rescue was associated with shootward localization of the fusion protein in epidermal and lateral cap cells of transformed *Arabidopsis* roots. Selection models suggest that the PIN2 gene underwent two rounds of selection during its evolution, including a first round of positive selection occurring early during the evolution of land plants, and a second round of purifying selection in seed plants, associated with fast gravitropic response.

This interesting comparative analysis of root gravitropism between multiple plant taxa is very novel, and provides potentially interesting clues on the mechanisms that evolved to allow fast and strong root gravitropism in land plants, as they evolved to progressively colonize dry lands. The manuscript is also well written. However, I have several concerns on the interpretation of the data.

First, the authors analyze the distribution of starch grains in the tip of roots coming from diverse land-plant taxa as well as green algae, showing that there is a progressive restriction of their distribution in the root apex (cap), away from the elongation zone, in plants that develop faster gravitropic responses. This information is aimed at supporting the conclusion that fast gravitropism correlates with a physical separation between sites of gravity sensing and curvature response, a process that also requires the involvement of an auxin transport machinery that translocates the auxin gradient generated across the cap in response to gravistimulation, toward the elongation zone, for curvature. However, many other cell types in plants accumulate starch grains without involving them in gravity sensing. One missing piece of information here, is whether the amyloplasts that appear along the elongation zone of ferns and lycophytes, for instance, have really anything to do with gravity sensing. For instance, do they sediment in response to plant reorientation within the gravity field? Importantly, do the amyloplasts present in the root apex of ferns sediment? If those function in gravity signal transduction, it should not be that relevant that more starch grains are also found in other regions of the root.

Another important piece of information that is missing from this analysis is the relative rate of organ growth between these diverse plant species. Because the authors made no attempt to quantify both the rate of gravitropic curvature and the rate of root growth for these plants, and figure 1 contains no scale bars, it is impossible to conclude whether the slower rate of gravitropism displayed by non-seed plants is also associated with slower rates of growth.

I think the latter point is important because colonization of dry lands required not only a strong root gravitropic response, but also an ability to grow faster to reach out for limited soil resources, and also alter growth patterns, independently from gravity, based on directional cues provided by gradients in water and nutrients to cite only a few. Hence, the strong focus on gravitropism as being the main selection target for plants colonizing dry lands may be overlooking a number of other processes that could have been selected for during this transition, and the traits analyzed here (starch accumulation and PIN2 protein distribution) may have evolved as a consequence of selection for other important parameters/growth behaviors that needed to be optimized for this new environment.

Considering the previous discussion, it seems odd that this research did actually not involve an analysis of clade-2 proteins, including PIN3, PIN4 and PIN7. These PIN proteins are expressed in the gravity-sensing statocytes of Arabidopsis roots (as nicely discovered by the authors' lab), and contribute to gravitropism through a rapid change in their distribution within the statocytes. It would seem quite appropriate to include a similar comparative analysis of their conservation/divergence, expression patterns and distribution between the taxa included in this analysis.

In their functional analysis of Arabidopsis PIN2 paralogs, the authors report that Arabidopsis PIN1 cannot rescue the gravitropism of pin2 mutant roots when expressed under the control of the PIN2 promoter, unlike PIN2 orthologs from gymnosperms. This is a very interesting piece of information. However, we already know from work done in the Friml laboratory that a phosphor-mimic mutant of PIN1 (only one amino acid change within the central hydrophilic domain compared to the wild type PIN1 protein!) is capable of rescuing the pin2 phenotype in a very similar assay, and properly localize in the lateral cap and epidermal cells of the meristem and elongation zone (Zhang et al., PNAS, 2010, not cited in this manuscript). Is this the only requirement for proper localization and function within the peripheral tissues of the root tip? If it is, it would really be useful to establish whether the gymnosperm and angiosperm PIN2 orthologs also carry this phosphorylation site. In other words, is this the only site to follow in selection studies? This manuscript should really include the results from this early publication in to consideration within the larger context of their current conclusions.

A few additional minor comments/suggestions follow:

- 1) It would be useful to point out in this manuscript that alternative mechanisms of gravity sensing also exist in plants, including the use of BaSO₄ -crystal-containing vacuoles as statoliths in some species such as Chara, as well as a secondary site of gravity sensing in the distal elongation zone of flowering-plant roots.
- 2) The 'apical-basal' nomenclature for cell polarity in roots should be avoided because it has been used differently from previously accepted rules in the past few years. Several root biologists have suggested using shootward vs rootward to define such polarity, precisely to avoid confusion.
- 3) Page 10, line 215. Please spell out HL where it appears first.
- 4) Figure-2 legend, last sentence. It seems like an oversimplification to think that all amyloplasts in roots are sensing gravity. In fact, this has not been tested for most of these species. Even a simple test of amyloplast sedimentation has not been carried out for some of these plants. Are the amyloplasts located at the bottom of the cells in vertical roots in all cases?
- 5) Figure 3. The roots in panels b-e are barely visible. These pictures should be replaced by better ones.
- 6) Figure 4, panel f. How many roots were analyzed for VGI quantification? Same for extended figure 2, panel c.
- 7) Legend to Extended Data Figure 1. The gravity vector cannot be shifted on earth (unless one uses a centrifuge to add an acceleration force). The plant orientation relative to the gravity vector was modified in this experiment.

Reviewer #2 (Remarks to the Author):

This manuscript describes evidence that angiosperms and gymnosperms share a fast root

gravitropism mechanisms, and have specific PIN proteins that may function in root gravitropism and which are able to adopt apical sub-cellular localisations in root epidermal cells.

There are some really exciting data here, which should lead to a good publication in due course. The analysis of gravitropism and gravi-sensing cells across land plants are particularly insightful, and the complementation and domain swap work is elegant and convincing.

However, some key additional experiments are needed to support the authors' conclusions. Furthermore, there are a number of major issues with the manuscript, particularly with regard to the evolutionary analyses and interpretations. These require re-structuring and re-writing of the manuscript, in addition to some additional analyses.

Major Issues:

1) Poor evolutionary framework

The authors' description and interpretation of evolution -- both of plants themselves, and of PIN proteins -- is lacking in care and accuracy. The title sums it up: "Evolution of efficient root gravitropism during conquest of land by plants" -- this is not what the manuscript actually shows.

In various places, this handling of evolutionary matters has major implications for the authors' interpretation of their data. See detailed comments below.

2) Low quality and unnecessary phylogenetic analysis

The authors present a new phylogeny of PIN proteins. However, there are already extensive analyses of PIN protein phylogeny with much higher and deeper sampling, and much higher resolution than the phylogeny presented here (e.g. Bennett et al, 2014). The phylogeny presented here gives different results to previous analyses, but there is no reason to believe this is a superior phylogeny, and plenty of reasons to believe it is not (i.e. limited sampling, no effort made to test the reliability of the reconstruction). There is no justification for using a new phylogeny over the existing ones. The authors either need to interpret their data in the light of existing phylogenies, or perform major new analyses that yield reliable new insights.

3) Are PIN2 and "PIN2-like" proteins closely related?

The authors heavily imply throughout the manuscript that the gymnosperm PtPIN-726 and PtPIN-100 proteins are 'PIN2-like' and form a sister clade to PIN2 proteins from angiosperms e.g. last results section, e.g. Extended data Figure 10; e.g. "These results suggest that the functional PIN2 crucial for the fast root gravitropism have originated in the most recent common ancestor of seed plants after the divergence from the basal vascular plant lineages."

However, even if the authors' new phylogeny was superior to existing ones, that is not actually what it shows. The tree actually shows that PIN2 is sister to a large clade of proteins that includes PtPIN-100 and PtPIN-726, but also includes PIN1, PIN11 and PtPIN-882. PtPIN-726 and PtPIN-100 are not sister to PIN2 proteins in this phylogeny, and they only appear to be adjacent to PIN2 proteins in the tree because the tree has been rotated to make them adjacent.

The gymnosperm sequences in question are from two separate previously described clades; PtPIN-726 is PtPINH and PtPIN-100 is PtPING. The overall suggestion of Bennett et al (2014) was that canonical PIN proteins have diversified independently in gymnosperms and angiosperms, and that PING/PINH are not co-orthologs of PIN2. However, there are some analyses in Bennett et al (particularly at the protein level) that suggest an affinity between PIN2 and PING/PINH (Bennett et al, 2014; Figure 3). So it is definitely possible that PING/PINH could be PIN2 co-orthologs. However, problematically for this manuscript, that is not what the phylogenetic evidence presented here (or elsewhere) shows.

Another point which needs addressed is that there is no specific structural similarity in the hydrophilic loop of PING/PINH and PIN2 to suggest they share a specific shared descent (Bennett et al, 2014; Figure 8). There is no clear reason why the HLs of PING/PINH and PIN2 should generate the same properties when the HLs of the other PINs do not.

In summary, on the basis of the current evidence, it is unclear whether the properties of PIN2 and PING/PINH are convergent, or whether they originate from shared descent.

Either the authors need more evidence, or they need to be much more careful about how they interpret their data in the manuscript.

4) No evidence that PING/PINH actually function in root gravitropism

For me, the most critical issue for the authors is the lack of any data showing the PING/PINH actually function in root gravitropism in gymnosperms. While it would be unreasonable to expect the authors to provide direct functional data that PING/PINH function in gravitropism, the authors could at least show that PING/PINH are expressed in gymnosperm root tips (e.g. qPCR) and that PING/PINH show apical localization in gymnosperm epidermal cells (e.g. immunolocalisation).

5) "Two-step functional innovation of PIN2"

The authors try to present the evolution of PIN2 as a two-step process. However, the first step (innovation in the TMDs) is common to all land plant PINs. There is no evidence that non-land plant PINs are auxin transporters, so this first step could be something as simple as evolving to transport auxin. It is only the second step that is actually an innovation creating PIN2-like proteins. This framework is unnecessary and misleading, and I suggest the authors remove it.

6) "Positive and purifying selections during PIN2 evolution"

The whole last result section adds nothing to manuscript. By definition, if a new gene function arises it must go through both positive and purifying selection. This section does not add any useful evidence that PIN2 has evolved a different function, and no one would seriously dispute that PIN2 does have a different function to other PINs. The section is also very difficult to read, and very heavy on technical details. It is also not associated with a primary figure. I strongly suggest the authors just remove this section.

Detailed comments on manuscript sections:

A) Title: "Evolution of efficient root gravitropism during conquest of land by plants" -- this is not what the manuscript shows. The manuscript shows that plants did not evolve fast root gravitropism for millions of years after they had conquered land. "Evolution of fast root gravitropism in seed plants" would be a more accurate title.

B) Introduction:

Line 30: "...among most important, the evolution of efficient root gravitropic response that allows roots to grow deep into the soil.". There is no evidence this was the case, and indeed, the authors' results suggest it is was actively not the case. This statement makes the false assumption that the only way roots can grow into the soil is gravitropically, but that is not the case.

Line 34/35: "and especially in the flowering plants, the root fully evolved into an organ to grow downwards along the gravity vector". No evidence that this is 'especially' in the flowering plants - clearly gymnosperms have gravitropic root growth too.

Line 46: "mediating by the PIN2 protein"  mediated

Line 53-56: "Answering this fundamental question would reveal how during plant evolutionary history root evolved to be such an efficient device to respond to the Earth gravity, and also provide an insight into how plants had adapted to the new, originally hostile environment to conquer the land." I agree with the first part of this sentence, but this work does not provide an insight into how plants conquered land.

C) Results section 1/Figure 1

Line 64: "The moss *Physcomitrella patens*, whose direct ancestor might be one of the earliest land plants, only evolved rhizoids"

This sentence is very confused - *Physcomitrella* is not an early land plant, and all plants are equally related to the earliest land plants  "Mosses, including the model *Physcomitrella patens*, have rhizoids but no true root"

Line 67: "The most primitive living vascular plants, the lycophyte *Selaginella moellendorffii* and the fern *Ceratopteris richardii*". *Selaginella* and *Ceratopteris* are not the most primitive living vascular plants. Neither are any extant vascular plants "primitive".  "Lycophytes and ferns have a true root, but the model lycophyte *Selaginella moellendorffii* and the model fern *Ceratopteris richardii* showed much slower gravitropism than the roots..."

Line 67: Persistent mis-spelling of *moellendorffii* as "moellendorffii":

Line 70 (and Line 90): "The basal seed plant". *Pinus taeda* is not a basal seed plant. It is just a seed plant.

D) Results section 2/Figure 2

Line 101: "epidemic cells"  epidermal cells

Line 107: "specifically accumulated specifically" - modify

E) Results section 3/Figure 3

Line 122 & elsewhere: persistent misuse of the term 'paralogous'.  "There are eight PIN genes in *Arabidopsis*"

Line 123 & Figure 3A: This is an NJ tree, so not a phylogenetic tree at all. What is the point of it? It does not correctly represent PIN protein evolution, but is used to talk about there being "three lineages" of PIN proteins. This is all incorrect, and the figure should be removed.

Line 123: "To determine which of the paralogous PINs evolved the ability to mediate the fast root gravitropism"  "To determine which of the PIN proteins can mediate fast root gravitropism"

Line 132-134: "confirming that of these paralogous PINs, only the PIN2 gene evolved to be a specific member of the PIN family to exclusively mediate the fast root gravitropism function in flowering plants." This sentence is a mess.  "confirming that only PIN2 can mediate fast root gravitropism in *Arabidopsis*"

F) Results section 4/Figure 3

As discussed above, the phylogenetic analysis presented in the first part of this section is pointless and adds nothing compared to previous studies. This just creates new problems.

Also, the new proteins identified in this section (e.g. CrPIN314 and CrPIN318, and all the gymnosperm proteins) should be named according to which of the existing clades they are in. e.g. PtPIN-100 PtPING

Line 149@ "in which PIN2 orthologs". None of these genes are PIN2 orthologs.

Line 151, 152: *Klebsormidium flaccidum* is actually *Klebsormidium nitens*. Also, there is a lack of evidence that KfPIN is actually an auxin transporter.

Line 154: MpPINZ, not MpPINW, is the only canonical PIN protein found in *Marchantia polymorpha*. MpPINW is highly non-canonical. If the authors did indeed use MpPINW, they need to re-do these experiments with MpPINZ.

Line 154: Also, *Marchantia polymorpha* is not "the earliest diverging land plant". It could be a representative of the earliest diverging land plants, but all recent analyses suggest that liverworts are not the earliest diverging land plants.

Line 160: There are five clades of PIN proteins in gymnosperms (Bennett et al, 2014). The authors identified 5 genes, one for each clade, not 4 clades.

Line 167-172 and Line 177-180: As discussed above, these results might be true, but that is not what the authors data actually show.

G) Results section 5/Figure 4

Line 183-185: This is a circular argument. The authors already assume that apical localisation is the "special functional property", because it is the only thing they test. They should just say "we

hypothesised that apical localisation was the innovation leading to fast gravitropism".

Line 195: "primitive living land plants" Not the correct phrasing.

Line 196: Needs more careful interpretation. The authors did not test every type of PIN protein from non-seed plants, and cannot exclude the possibility at least some of them can apically localise in Arabidopsis roots. Testing a few examples is not sufficient evidence.

H) Results section 6/Extended Data 10

As discussed above, this section adds very little to the manuscript.

I) Discussion/Figure 5

Lines 310-324: This last section of the discussion contains a 'just so' story that makes the authors results seem very exciting in terms of land plant evolution. However, there is no evidence cited to support any of this model. Fast gravitropism is not a pre-requisite for deep rooting, which could theoretically be achieved just as well by slow gravitropism, or by hydrotropism. Is there evidence that ferns do not root deeply anyway? Deep rooting is not a necessary pre-requisite for colonization of dry environments either; plenty of liverworts and mosses live in dry environments. Being tall is not an exclusive property of seed plants, and does not require deep roots anyway - the rooting systems of many trees are rather shallow. Furthermore, being tall is not required for wind pollination.

The authors should remove this whole section, and write something that can actually be justified in terms of land plant evolution.

Reviewers' comments:

Reviewer #1 (Remarks to the Author):

The manuscript by Yuzhou Zhang and collaborators (NCOMMS-19-04573-T: Evolution of Efficient Root Gravitropism During Conquest of Land by Plants) reports on investigations aimed at better understanding the evolution of the machinery associated with efficient, fast root gravitropism in flowering plants. The authors note that the specific accumulation of starch grains in the root apex, physically separated from the graviresponding elongation zone, appeared coincidentally with fast root gravitropism in land plants. They point out that the need to transport auxin from the site of sensing to the site of curvature response (the elongation zone) was fulfilled by appearance of a shootward-localized efflux carrier, PIN2. They develop a phylogenetic analysis of plant PIN proteins, which shows the existence of clearly separated clades containing PIN2 proteins in the flowering plants (named clade 4) and another one containing gymnosperm PIN2-like proteins (clade 3). Using transformation–rescue experiments of *Arabidopsis pin2* knockout plants, they demonstrate that angiosperm PIN2 and gymnosperm PIN2-like proteins, are capable of fully rescuing the gravitropic defect of the mutant, whereas other *Arabidopsis* paralogous PINs belonging to different clades (PIN1, for instance), or PIN proteins found in non-seed plants, cannot rescue the *Arabidopsis* mutant. Furthermore, replacing the central hydrophilic loop of these heterologous proteins with that from *Arabidopsis* PIN2 led to rescue as long as the initial protein derived from a land plant (the fusion protein carrying the green algae transmembrane domains was not able to rescue the mutant phenotype in *Arabidopsis*). Similarly, replacing the central hydrophilic loop from these proteins by those from gymnosperms, allowed rescue of the gravitropic response in *pin2* mutant roots. In all these transformation-rescue experiments, gravitropism rescue was associated with shootward localization of the fusion protein in epidermal and lateral cap cells of transformed *Arabidopsis* roots. Selection models suggest that the PIN2 gene underwent two rounds of selection during its evolution, including a first round of positive selection occurring early during the evolution of land plants, and a second round of purifying selection in seed plants, associated with fast gravitropic response.

This interesting comparative analysis of root gravitropism between multiple plant taxa is very novel, and provides potentially interesting clues on the mechanisms that evolved to allow fast and strong root gravitropism in land plants, as they evolved to progressively colonize dry lands.

The manuscript is also well written. However, I have several concerns on the interpretation of the data.

RESPONSE: We thanks for the reviewer 1's positive comments about our manuscripts and about the novelty of the results reported.

First, the authors analyze the distribution of starch grains in the tip of roots coming from diverse land-plant taxa as well as green algae, showing that there is a progressive restriction of their distribution in the root apex (cap), away from the elongation zone, in plants that develop faster gravitropic responses. This information is aimed at supporting the conclusion that fast gravitropism correlates with a physical separation between sites of gravity sensing and curvature response, a process that also requires the involvement of an auxin transport machinery that translocates the auxin gradient generated across the cap in response to gravistimulation, toward the elongation zone, for curvature. However, many other cell types in plants accumulate starch grains without involving them in gravity sensing. One missing piece of information here, is whether the amyloplasts that appear along the elongation zone of ferns and lycophytes, for instance, have really anything to do with gravity sensing. For instance, do they sediment in response to plant reorientation within the gravity field? Importantly, do the amyloplasts present in the root apex of ferns sediment? If those function in gravity signal transduction, it should not be that relevant that more starch grains are also found in other regions of the root.

RESPONSE: Thanks for this excellent suggestion. We addressed this point by examining the sedimentation of amyloplasts in both the lycophyte and fern roots after 180° reorientation of the seedlings (See new **Supplementary Fig. 5**).

In contrast to the fast sedimentation of the amyloplasts in the root apex of the *Arabidopsis thaliana*, the amyloplasts in the root cap/elongation zone (EZ) of the fern *C. richardii* and in the EZ of lycophyte *S. Moellendorffii* clearly showed a random localization without any obvious sedimentation before and after the gravistimulation. This suggests that the amyloplasts in the roots of these basal vascular plants do not serve as the statoliths and might be not involved in the gravity sensing. The Result section is revised accordingly (Line 116-124, Page 6).

Another important piece of information that is missing from this analysis is the relative rate of organ growth between these diverse plant species. Because the authors made no attempt to quantify both the rate of gravitropic curvature and the rate of root growth for these plants, and

figure 1 contains no scale bars, it is impossible to conclude whether the slower rate of gravitropism displayed by non-seed plants is also associated with slower rates of growth.

RESPONSE: A very valid point. **a)** The scale bars are now provided in Fig. 1. **b)** The growth rate of these diverse plant species with different time points (0h, 3h, 6h, 9h, 12h) is now provided (See new **Supplementary Fig. 2a**), which shows indeed a slower growth rate of the lycophyte/fern roots as compared with that of the seed plant roots. Nonetheless, not that much slower to explain the much less efficient bending kinetics. **c)** In addition, to minimize the effect of the growth rate on the evaluation of the root gravitropism, we now compare the rhizoid/roots with approximately the same additional elongation (~2 mm) after the gravistimulation to evaluate the Vertical Growth Index (VGI) (**Supplementary Fig. 2b, 2c**). This further confirmed the slower root gravitropism of these basal vascular plant species (lycophyte and fern) as compared to the seed plants. The main text is also revised accordingly (Line 75-80).

I think the latter point is important because colonization of dry lands required not only a strong root gravitropic response, but also an ability to grow faster to reach out for limited soil resources, and also alter growth patterns, independently from gravity, based on directional cues provided by gradients in water and nutrients to cite only a few. Hence, the strong focus on gravitropism as being the main selection target for plants colonizing dry lands may be overlooking a number of other processes that could have been selected for during this transition, and the traits analyzed here (starch accumulation and PIN2 protein distribution) may have evolved as a consequence of selection for other important parameters/growth behaviors that needed to be optimized for this new environment.

RESPONSE: Agreed! We have taken this comment and a similar reviewer 2's comment into consideration to revise our manuscript and re-interpret our results. Now we mainly describe the evolution of the fast gravitropism in seed plants, and treat it as one of the selected traits/consequences during the plant colonization of the dry land, rather than the main selection target for plants colonizing the dry land. The revisions are as follows: **1)** the title is now revised to "Evolution of fast root gravitropism in seed plants after conquest of land". **2)** The last sentence in the Abstract is revised to read: "as one of the important adaptations to dry land in seed plants". **3)** In the Discussion section, the last paragraph is completely rewritten to describe in a more balanced way the fast gravitropism as one of the traits/consequences for seed plant adaptation to dry land. **4)** The model in the Fig. 5 is also now slightly modified, which we replaced

the statement of the “Fast root gravitropism for plant adaptation to the dry land” with the “Fast root gravitropism”, and the legend was also rewritten to avoid misleading statements.

All of the other revised places in the main text are also highlighted and can be easily tracked.

Considering the previous discussion, it seems odd that this research did actually not involve an analysis of clade-2 proteins, including PIN3, PIN4 and PIN7. These PIN proteins are expressed in the gravity-sensing statocytes of *Arabidopsis* roots (as nicely discovered by the authors' lab), and contribute to gravitropism through a rapid change in their distribution within the statocytes. It would seem quite appropriate to include a similar comparative analysis of their conservation/divergence, expression patterns and distribution between the taxa included in this analysis.

RESPONSE: Thanks for this suggestion. Now we revise our manuscript to include the clade-2 (PIN3 clade) in the evolutionary analysis and discuss it accordingly (Line 312-323, Page 14). Interestingly, according to the phylogenetic tree (the Supplementary Fig. 8), besides in the flowering plants, the orthologs of PIN3 clade (PINE) could also be found in the gymnosperms, but this clade is clearly absent in the non-seed plant species (ferns/lycophytes), which is similar to the evolutionary analysis of the PIN2 clade. These results are also congruent with our observation that the fast root gravitropism evolved in seed plants rather than in the non-seed plants, suggesting the evolution of the PIN3/PINE clade may also play an important role in the origination of the fast root gravitropism in seed plants.

In their functional analysis of *Arabidopsis* PIN2 paralogs, the authors report that *Arabidopsis* PIN1 cannot rescue the gravitropism of *pin2* mutant roots when expressed under the control of the PIN2 promoter, unlike PIN2 orthologs from gymnosperms. This is a very interesting piece of information. However, we already know from work done in the Friml laboratory that a phosphor-mimic mutant of PIN1 (only one amino acid change within the central hydrophilic domain compared to the wild type PIN1 protein!) is capable of rescuing the *pin2* phenotype in a very similar assay, and properly localize in the lateral cap and epidermal cells of the meristem and elongation zone (Zhang et al., PNAS, 2010, not cited in this manuscript). Is this the only requirement for proper localization and function within the peripheral tissues of the root tip? If it is, it would really be useful to establish whether the gymnosperm and angiosperm PIN2 orthologs also carry this phosphorylation site. In other words, is this the only site to follow in

selection studies? This manuscript should really include the results from this early publication in to consideration within the larger context of their current conclusions.

RESPONSE: Actually, in the published paper (Zhang et al., PNAS, 2010), two amino acids in the hydrophilic loop of PIN1 protein were artificially modified to the constitutive phospho-mimic. For these two amino acids identified, one (Ser) is missing and the other one (Thr) is conserved in PIN2. This suggests that the phosphorylation of PIN2 for its apical localization depends on other sites within the PIN2 protein. According to another publication (Dhonukshe et al., Development, 2010), there are other three key phosphorylation sites in *Arabidopsis* PIN2 protein, which were demonstrated to be crucial for the AtPIN2 apical subcellular localization and its function in root gravitropism. So we used these critical sites for the sequence analysis.

The sequence alignment (denoted by the new **Supplementary Fig. 13**) revealed that besides the *Arabidopsis* AtPIN2, both the other angiosperm PIN2 and its orthologous gymnosperm PINH/G carried these three conserved phosphorylation sites suggesting that these phosphorylation sites in the PIN2 orthologs have been evolutionarily conserved within the seed plants during the evolution. We also added some sentences to describe it (Line 223-229 Page 10).

A few additional minor comments/suggestions follow:

1) It would be useful to point out in this manuscript that alternative mechanisms of gravity sensing also exist in plants, including the use of BaSO₄ –crystal-containing vacuoles as statoliths in some species such as *Chara*, as well as a secondary site of gravity sensing in the distal elongation zone of flowering-plant roots.

RESPONSE: These information are now provided in the revised manuscript (Line 41-43 and line 52-54 in the Page 3).

2) The ‘apical-basal’ nomenclature for cell polarity in roots should be avoided because it has been used differently from previously accepted rules in the past few years. Several root biologists have suggested using shootward vs rootward to define such polarity, precisely to avoid confusion.

RESPONSE: We agree that the original physiological nomenclature, especially the one involving terms acropetal and basipetal is confusing and therefore we fully agree to replace these term with shootward and rootward. However, there was never a confusion in the cell biological or developmental biology community, which derived the terms denoting side of cells from its inheritance from the early embryo. Currently, it is still more practical to use these terms

for cell biological questions and majority of the community does so. Not least because of the parallels to the apical/basal nomenclature in non-plant cell biology. Therefore, we would like to keep it as usually used but clearly define the term at the beginning.

3) Page 10, line 215. Please spell out HL where it appears first.

RESPONSE: Corrected. Now in the line 241, Page 11.

4) Figure-2 legend, last sentence. It seems like an oversimplification to think that all amyloplasts in roots are sensing gravity. In fact, this has not been tested for most of these species. Even a simple test of amyloplast sedimentation has not been carried out for some of these plants. Are the amyloplasts located at the bottom of the cells in vertical roots in all cases?

RESPONSE: Thanks for pointing it out. It is now corrected to avoid this misleading description (According to the Supplementary Fig. 5) by replacing the 'gravity-sensing amyloplasts' with the 'amyloplasts' (Page 26, line 584).

5) Figure 3. The roots in panels b-e are barely visible. These pictures should be replaced by better ones.

RESPONSE: The clear pictures were re-taken to replace them. In addition, we also removed the Fig. 3a according the Reviewer 2's comment and then enlarged these panels (now the Fig. 3a-d) to make it more visible.

6) Figure 4, panel f. How many roots were analyzed for VGI quantification? Same for extended figure 2, panel c.

RESPONSE: The number of the roots (denoted by n) used for analysis is now provided in the Fig. 4f, Supplementary Fig. 3c (the Extended Data Fig. 2 in the original version).

7) Legend to Extended Data Figure 1. The gravity vector cannot be shifted on earth (unless one uses a centrifuge to add an acceleration force). The plant orientation relative to the gravity vector was modified in this experiment.

RESPONSE: Agreed and thanks for pointing it out, as you suggested, now we use the 'plant orientation' to replace the phrase 'gravity vector'.

Reviewer #2 (Remarks to the Author):

This manuscript describes evidence that angiosperms and gymnosperms share a fast root gravitropism mechanisms, and have specific PIN proteins that may function in root gravitropism and which are able to adopt apical sub-cellular localisations in root epidermal cells.

There are some really exciting data here, which should lead to a good publication in due course. The analysis of gravitropism and gravi-sensing cells across land plants are particularly insightful, and the complementation and domain swap work is elegant and convincing.

RESPONSE: We much appreciate the reviewer 2's encouraging comments about our manuscripts.

However, some key additional experiments are needed to support the authors' conclusions. Furthermore, there are a number of major issues with the manuscript, particularly with regard to the evolutionary analyses and interpretations. These require re-structuring and re-writing of the manuscript, in addition to some additional analyses.

Major Issues:

1) Poor evolutionary framework

The authors' description and interpretation of evolution -- both of plants themselves, and of PIN proteins -- is lacking in care and accuracy. The title sums it up: "Evolution of efficient root gravitropism during conquest of land by plants" -- this is not what the manuscript actually shows.

RESPONSE: The title is now revised to "Evolution of fast root gravitropism in seed plants after conquest of land". We hope that this is o.k.

In various places, this handling of evolutionary matters has major implications for the authors' interpretation of their data. See detailed comments below.

2) Low quality and unnecessary phylogenetic analysis

The authors present a new phylogeny of PIN proteins. However, there are already extensive analyses of PIN protein phylogeny with much higher and deeper sampling, and much higher resolution than the phylogeny presented here (e.g. Bennett et al, 2014). The phylogeny presented here gives different results to previous analyses, but there is no reason to believe this is a superior phylogeny, and plenty of reasons to believe it is not (i.e. limited sampling, no effort made to test the reliability of the reconstruction). There is no justification for using a new

phylogeny over the existing ones. The authors either need to interpret their data in the light of existing phylogenies, or perform major new analyses that yield reliable new insights.

RESPONSE: Agreed! The PIN phylogeny reported by Bennett *et al* (*Mol. Bio. Evol.* 2014) has been the most comprehensive one with the highest quality so far, so we used this literature as the golden standard to carefully interpret the evolutionary relationship of these PIN members throughout the manuscript. Notably, the interpretations of this and our phylogeny does not differ much. We made textual revisions accordingly: (a) line156-159, Page 8, we completely rewrote it again in light of the previous phylogenetic analysis and also used the "According to the comprehensive PIN phylogeny by Bennett, *et al*⁴²" to replace the original statement that "According to the tree" by us. (b) We renamed all the *PIN* genes in the ferns and gymnosperms according to this literature. (c) The other numerous changes could be found in those revisions according to the comments #3 and #5. All the changes have been highlighted and could be tracked in this revised manuscripts.

3) Are PIN2 and "PIN2-like" proteins closely related?

The authors heavily imply throughout the manuscript that the gymnosperm PtPIN-726 and PtPIN-100 proteins are 'PIN2-like' and form a sister clade to PIN2 proteins from angiosperms e.g. last results section, (Removed according to the comment #6)
e.g. Extended data Figure 10; (Removed according to the comment #6)
e.g. "These results suggest that the functional PIN2 crucial for the fast root gravitropism have originated in the most recent common ancestor of seed plants after the divergence from the basal vascular plant lineages."

RESPONSE: Now we replace the 'functional PIN2' with 'functional PIN' and delete the 'most common ancestor' as well (now Line 186-189, Page 9), to avoid the misleading that the gymnosperm PINH/G (PIN726/100 in the old version) and flowering plant PIN2 originated from a shared descent.

However, even if the authors' new phylogeny was superior to existing ones, that is not actually what it shows. The tree actually shows that PIN2 is sister to a large clade of proteins that includes PtPIN-100 and PtPIN-726, but also includes PIN1, PIN11 and PtPIN-882. PtPIN-726 and PtPIN-100 are not sister to PIN2 proteins in this phylogeny, and they only appear to be adjacent to PIN2 proteins in the tree because the tree has been rotated to make them adjacent.

The gymnosperm sequences in question are from two separate previously described clades; PtPIN-726 is PtPINH and PtPIN-100 is PtPING. The overall suggestion of Bennett et al (2014) was that canonical PIN proteins have diversified independently in gymnosperms and angiosperms, and that PING/PINH are not co-orthologs of PIN2. However, there are some analyses in Bennett et al (particularly at the protein level) that suggest an affinity between PIN2 and PING/PINH (Bennett et al, 2014; Figure 3). So it is definitely possible that PING/PINH could be PIN2 co-orthologs. However, problematically for this manuscript, that is not what the phylogenetic evidence presented here (or elsewhere) shows.

Another point which needs addressed is that there is no specific structural similarity in the hydrophilic loop of PING/PINH and PIN2 to suggest they share a specific shared descent (Bennett et al, 2014; Figure 8). There is no clear reason why the HLs of PING/PINH and PIN2 should generate the same properties when the HLs of the other PINs do not.

In summary, on the basis of the current evidence, it is unclear whether the properties of PIN2 and PING/PINH are convergent, or whether they originate from shared descent.

Either the authors need more evidence, or they need to be much more careful about how they interpret their data in the manuscript.

RESPONSE: We checked it again, and fully agree that the evidences collected so far is unable to decide whether the gymnosperms PING/PINH and flowering plants PIN2 have evolved independently or originated from a shared descent. Now we take into account of the both possibilities to carefully revise our manuscript. The changes are as follows.

1) According to the literature by Bennett, 2014 *et al.*, we used PtPINH and PtPING to replace the original names 'PtPIN-726' and 'PtPIN-100', respectively. The names of the other gymnosperm PINs is also corrected accordingly.

2) We avoid using the word 'PIN2-like' or 'PIN2 orthologs' throughout the whole manuscripts, which denoted the PtPING/H in the previous version (e. g., the changes in Line 161-162, Line 190-191, line 198). Meanwhile, according to the comment #2, we use the PIN phylogeny by *Bennett, et al* to interpret the relationship of the PIN family members.

3) In the RESULT section, we rewrote the sentence (the line 186-189, page 9) mentioned at the beginning of this comment, and also revised the other parts of the main text to avoid the misleading that PtPING/H seems to have a shared descent with flowering plant PIN2.

4) We felt that these important information that the reviewer mentioned above should be pointed out, so we added one paragraph in the DISCUSSION section to include the two possible originations/evolutions of the PIN2 and PING/PINH (line 301-311, Page 13).

All the changes have been highlighted in the revised manuscripts to facilitate your reading. Many thanks for this excellent comment.

4) No evidence that PING/PINH actually function in root gravitropism. For me, the most critical issue for the authors is the lack of any data showing the PING/PINH actually function in root gravitropism in gymnosperms. While it would be unreasonable to expect the authors to provide direct functional data that PING/PINH function in gravitropism, the authors could at least show that PING/PINH are expressed in gymnosperm root tips (e.g. qPCR) and that PING/PINH show apical localization in gymnosperm epidermal cells (e.g. immunolocalisation).

RESPONSE: Thanks for your understanding. The new data (the new **Supplementary Fig. 10**) are now provided to show that the two gymnosperm PIN genes PtPING/PtPINH, especially the PtPING, are strongly expressed in the root tip of *P. taeda* as compared with their expression level in the other part of the root or in the shoot by RT-PCR and qPCR (Supplementary Fig. 10a and b), implying the important role of the PtPING/PINH in mediating the auxin transport in the root tip of *P. taeda*.

2) We apologize that the time given to revise the manuscript is not sufficient to generate and verify the antibody of PtPING/H for immunolocalisation, and also the potential technical difficulty to perform immunolocalization with pine tree's root. As an alternative, we used the auxin transport assay with the radioactive ^3H -labelled IAA (^3H -IAA) to test the shootward auxin transport activity in the gymnosperm root tip. In contrast to much less efficient shootward transport in ferns, we found efficient; NPA-sensitive shootward transport in gymnosperm (Supplementary Fig. 10d, e). This does not only prove our original hypothesis than only seed plants evolved shootward auxin transport but also in combination with the strong expression of the PtPING/H in the root tip (Supplementary Fig. 10b) and shootward subcellular localization of the PtPING/H when expressed in *Arabidopsis* (Fig. 3w, x), implies that the PtPING/H play an

important role in this process. We also described these results in our revised manuscript. Line 178-189, Page 8-9; Line 221, Page 10.

5) "Two-step functional innovation of PIN2"

The authors try to present the evolution of PIN2 as a two-step process. However, the first step (innovation in the TMDs) is common to all land plant PINs. There is no evidence that non-land plant PINs are auxin transporters, so this first step could be something as simple as evolving to transport auxin. It is only the second step that is actually an innovation creating PIN2-like proteins. This framework is unnecessary and misleading, and I suggest the authors remove it.

RESPONSE: Indeed, until now, there is really no publication to show that the non-land plant PINs are auxin transporters. However, according to our unpublished data (another manuscript currently close to resubmission), it clearly shown that the non-land plant *Klebsormidium flaccidum* PIN (KfPIN) is an auxin transporter (based on phenotypes in land plant models, transport assays in cultured cells and in *Xenopus* oocytes). We refer now to this work in line 165-166 page 8 to clarify it. Meanwhile, we also rewrote the manuscript to remove this misleading statement "Two-step functional innovation" throughout the manuscript as suggested.

6) "Positive and purifying selections during PIN2 evolution"

The whole last result section adds nothing to manuscript. By definition, if a new gene function arises it must go through both positive and purifying selection. This section does not add any useful evidence that PIN2 has evolved a different function, and no one would seriously dispute that PIN2 does have a different function to other PINs. The section is also very difficult to read, and very heavy on technical details. It is also not associated with a primary figure. I strongly suggest the authors just remove this section.

RESPONSE: Agreed. We moved this section to Supplements (Page 18-19 in the Supplementary Information).

Detailed comments on manuscript sections:

A) Title: "Evolution of efficient root gravitropism during conquest of land by plants" -- this is not what the manuscript shows. The manuscript shows that plants did not evolve fast root gravitropism for millions of years after they had conquered land. "Evolution of fast root gravitropism in seed plants" would be a more accurate title.

RESPONSE: Now corrected accordingly.

B) Introduction:

Line 30: "...among most important, the evolution of efficient root gravitropic response that allows roots to grow deep into the soil.". There is no evidence this was the case, and indeed, the authors' results suggest it is was actively not the case. This statement makes the false assumption that the only way roots can grow into the soil is gravitropically, but that is not the case.

Line 34/35: "and especially in the flowering plants, the root fully evolved into an organ to grow downwards along the gravity vector". No evidence that this is 'especially' in the flowering plants - clearly gymnosperms have gravitropic root growth too.

RESPONSE: Many thanks for your correction, we now correct the "Among most important" to "among them" (Line 31), and also remove the word 'especially' (Line 35).

Line 46: "mediating by the PIN2 protein"  mediated

RESPONSE: Line 49, now corrected.

Line 53-56: "Answering this fundamental question would reveal how during plant evolutionary history root evolved to be such an efficient device to respond to the Earth gravity, and also provide an insight into how plants had adapted to the new, originally hostile environment to conquer the land." I agree with the first part of this sentence, but this work does not provide an insight into how plants conquered land.

RESPONSE: Agreed and removed the last sentence (Line 60).

C) Results section 1/Figure 1

Line 64: "The moss *Physcomitrella patens*, whose direct ancestor might be one of the earliest land plants, only evolved rhizoids"

This sentence is very confused - *Physcomitrella* is not an early land plant, and all plants are equally related to the earliest land plants  "Mosses, including the model *Physcomitrella patens*, have rhizoids but no true root"

RESPONSE: Corrected it as suggested (now Line 67). Thanks!

Line 67: "The most primitive living vascular plants, the lycophyte *Selaginella moellendorffii* and the fern *Ceratopteris richardii*". *Selaginella* and *Ceratopteris* are not the most primitive living vascular plants. Neither are any extant vascular plants "primitive".  "Lycophytes and ferns

have a true root, but the model lycophyte *Selaginella moellendorffii* and the model fern *Ceratopteris richardii* showed much slower gravitropism than the roots..."

RESPONSE: Now corrected accordingly (now Line 70-71).

Line 67: Persistent mis-spelling of *moellendorffii* as "moellendoriffii":

RESPONSE: We have checked throughout the manuscripts and corrected all these misspellings.

Line 70 (and Line 90): "The basal seed plant". *Pinus taeda* is not a basal seed plant. It is just a seed plant.

RESPONSE: Now corrected as suggested (Line 73 and Line 97).

D) Results section 2/Figure 2

Line 101: "epidemic cells"  epidermal cells

RESPONSE: Now corrected as suggested (Line 108).

Line 107: "specificical accumulated specifically" – modify

RESPONSE: Now corrected to "specifically accumulated" (Line 113).

E) Results section 3/Figure 3

Line 122 & elsewhere: persistent misuse of the term 'paralogous'.  "There are eight PIN genes in *Arabidopsis*"

RESPONSE: Line 137 & elsewhere, now removed the word 'paralogous'.

Line 123 & Figure 3A: This is an NJ tree, so not a phylogenetic tree at all. What is the point of it? It does not correctly represent PIN protein evolution, but is used to talk about there being "three lineages" of PIN proteins. This is all incorrect, and the figure should be removed.

RESPONSE: Agreed and corrected (Line 137-138). Now the Fig. 3a is also removed as suggested.

Line 123: "To determine which of the paralogous PINs evolved the ability to mediate the fast root gravitropism"  "To determine which of the PIN proteins can mediate fast root gravitropism"

RESPONSE: Now corrected (Line 141).

Line 132-134: "confirming that of these paralogous PINs, only the PIN2 gene evolved to be a specific member of the PIN family to exclusively mediate the fast root gravitropism function in flowering plants." This sentence is a mess.  "confirming that only PIN2 can mediate fast root gravitropism in Arabidopsis"

RESPONSE: Now corrected as suggested (Line 146-147).

F) Results section 4/Figure 3

As discussed above, the phylogenetic analysis presented in the first part of this section is pointless and adds nothing compared to previous studies. This just creates new problems. Also, the new proteins identified in this section (e.g. CrPIN314 and CrPIN318, and all the gymnosperm proteins) should be named according to which of the existing clades they are in. e.g. PtPIN-100 PtPING

RESPONSE: We corrected all the names of the *PIN* genes from the ferns and gymnosperms according to the publication from *Bennett et al (Mol. Bio. Evol. 2014)*.

Line 149@ "in which PIN2 orthologs". None of these genes are PIN2 orthologs.

RESPONSE: Now we remove this misleading statement and replace it with the 'with *PIN* genes from various representative plant lineages' (Now Line 161-162).

Line 151, 152: *Klebsormidium flaccidum* is actually *Klebsormidium nitens*. Also, there is a lack of evidence that KfPIN is actually an auxin transporter.

RESPONSE: Indeed, *Klebsormidium flaccidum* (NIES-2285) used for the genome project by Hori *et al* (2014) was re-identified as *Klebsormidium nitens*. However, our KfPIN sequence is obtained from the unpublished transcriptome database provided by E. D. Cooper and C. F. Delwiche with the strain from UTEX culture collection (UTEX strain #321), which is different from the published one. Sorry for the missing information, now we point it out at the Method section (Line 341). Meanwhile, we also point out that the non-land plant *Klebsormidium flaccidum* PIN (KfPIN) is an auxin transporter (Line 165) as the revision made according to the comment #5.

Line 154: MpPINZ, not MpPINW, is the only canonical PIN protein found in *Marchantia polymorpha*. MpPINW is highly non-canonical. If the authors did indeed use MpPINW, they need to re-do these experiments with MpPINZ.

RESPONSE: Thanks for pointing it out. We really used the wrong name MpPINW to denote this canonical MpPINZ. Now corrected it throughout our manuscript (Line 166 and elsewhere).

Line 154: Also, *Marchantia polymorpha* is not "the earliest diverging land plant". It could be a representative of the earliest diverging land plants, but all recent analyses suggest that liverworts are not the earliest diverging land plants.

RESPONSE: Now corrected and we denote it as 'a probable representative of the earliest diverging land plants' (Line 167).

Line 160: There are five clades of PIN proteins in gymnosperms (Bennett et al, 2014). The authors identified 5 genes, one for each clade, not 4 clades.

RESPONSE: Now corrected accordingly (Line 173-174).

Line 167-172 and Line 177-180: As discussed above, these results might be true, but that is not what the authors data actually show.

RESPONSE: We have corrected it according to the comment #3. Now, Line 186-191 and Line 198: we rewrote these sentences to remove the term 'PIN2-like' and also revised them to avoid the misleading that the PtPING/H seems to be the co-orthologs of the PIN2.

G) Results section 5/Figure 4

Line 183-185: This is a circular argument. The authors already assume that apical localisation is the "special functional property", because it is the only thing they test. They should just say "we hypothesised that apical localisation was the innovation leading to fast gravitropism".

RESPONSE: Now line 203-204, we rewrote it as the reviewer suggested.

Line 195: "primitive living land plants" Not the correct phrasing.

RESPONSE: Many thanks for pointing it out. Taking into account the comment below, we rewrote this sentence with the phrasing 'primitive living land plants' removed (Line 212-214).

Line 196: Needs more careful interpretation. The authors did not test every type of PIN protein from non-seed plants, and cannot exclude the possibility at least some of them can apically localise in *Arabidopsis* roots. Testing a few examples is not sufficient evidence.

RESPONSE: Now we only use the '*P. patens*' to make the description much more precise (Line 212-214).

H) Results section 6/Extended Data 10

As discussed above, this section adds very little to the manuscript.

RESPONSE: Agreed and now fully removed it according to the comment #6.

I) Discussion/Figure 5

Lines 310-324: This last section of the discussion contains a 'just so' story that makes the authors results seem very exciting in terms of land plant evolution. However, there is no evidence cited to support any of this model. Fast gravitropism is not a pre-requisite for deep rooting, which could theoretically be achieved just as well by slow gravitropism, or by hydrotropism. Is there evidence that ferns do not root deeply anyway? Deep rooting is not a necessary pre-requisite for colonization of dry environments either; plenty of liverworts and mosses live in dry environments. Being tall is not an exclusive property of seed plants, and does not require deep roots anyway - the rooting systems of many trees are rather shallow. Furthermore, being tall is not required for wind pollination.

The authors should remove this whole section, and write something that can actually be justified in terms of land plant evolution.

RESPONSE: We have completely rewritten the last paragraph (now Line 324-333, Page 14) to mainly describe the fast root gravitropism evolved in the seed plants, and simply treated it as one of the selected traits/consequences during the plant colonization of the dry land, rather than the most critical feature or the main selection target for plants colonizing the dry land as described in the previous version. Moreover, we also added one paragraph in the Discussion section to include the both two possible originations of the PIN2 and PING/PINH in light of the comment #3 (Line 301-311, Page 13).

Thank you so much for your careful reading and professional and very useful comments.

REVIEWERS' COMMENTS:

Reviewer #1 (Remarks to the Author):

This revised version of the manuscript by Yuzhou Zhang and collaborators (NCOMMS-19-04573A: "Evolution of Fast Root Gravitropism in Seed Plants after Conquest of Land") takes into account most of the comments and suggestions made by the reviewers on the previous draft. In particular, the authors provide new data that show a lack of amyloplast sedimentation in lycophyte and fern roots. They take into account the relative root growth rates in their comparative analyses of root gravitropism between species, and provide a more nuanced view on the evolutionary process, acknowledging the possible involvement of other selective forces. They document the conservation of specific phosphorylation-target sites known to contribute to polar localization of the PIN2 and PINH/G proteins in flowering plants and gymnosperms, and include a new paragraph discussing the important contribution of PIN3, PIN4 and PIN7 proteins in fast root gravitropism in flowering plants and gymnosperms (along with their absence in other plant clades displaying evidence of slower root gravitropism). They also provide a better phylogenetic analysis of PIN proteins that is more compatible with previously published trees. Overall, the changes made to this manuscript have substantially enhanced its quality. As pointed out in my review of the first draft, the data presented here are novel and should be of a broad interest to the scientific community. However, I still have a few concerns about this work, which need more attention:

- 1) The analysis of amyloplast sedimentation in root cells of lycophytes and ferns revealed an inability to sediment in these species under treatments that normally promote their sedimentation in flowering plant columella cells. From this observation, the authors conclude: "These results strongly suggest that, unlike in the flowering plant roots, the amyloplasts in the roots of these basal vascular plants did not evolve to serve as the statolith to participate in gravity sensing". I would recommend more caution in the interpretation of the data. Indeed, independently of whether they sediment or not, these amyloplasts still contain starch and, therefore, are likely to be denser than the cytoplasm. Even if they are tightly anchored to the cytoskeleton, they still have the potential of exerting large, gravity dependent forces onto the cytoskeleton, thereby transmitting information related to the gravity vector to the rest of the cell. If this occurs, the mechanism of gravity sensing would probably be somewhat different from the one used by the columella cells of the cap in flowering plants. However, these non-sedimenting amyloplasts would still contribute to gravity sensing (potentially, at least).
- 2) The discussion on the role of PIN3 in root gravitropism focuses on reinforcing the conclusion that "this PIN clade also evolved to facilitate the fast root gravitropism of the seed plants after their divergence from the fern lineage" (lines 321-323). This conclusion seems to completely miss the fact that this clade is missing not only in the non-seed plants, but also in the monocots. This observation should be incorporated in the evolutionary model discussed in this manuscript.
- 3) I continue to strongly believe that the apical-basal nomenclature is highly confusing nowadays when applied to plant biology as it has been used to define opposite sides of the cells / structures in roots by different authors. I would argue that a paper that is likely to target the large and diverse readership of a journal of the stature of Nature Communications should attempt to eliminate any terms that might confuse a section of the readership.
- 4) Most of the quantitative data presented in this manuscript lack any statistical analysis of significance. For instance, statistical analyses (and corresponding symbols) should be added to Figure 3 panel m, Figure 4 panel f, Suppl. Figure 2 (panels a and c), Suppl. Figure 6, panel b, Suppl. Figure 9, panel B, and Suppl. Figure 10, panels c and e.
- 5) The title of Suppl. Figure 4 is misleading. Indeed, this figure shows only the location of amyloplasts in root tips, but it does not address the actual site of gravity sensing in these roots. This would require additional functional analyses that have not been carried out in this work.

Reviewer #2 (Remarks to the Author):

I previously reviewed this manuscript, and thought there were very exciting data, but that the interpretation of the data needed to be improved. The authors have addressed all my major concerns, and I feel this is now a much more coherent manuscript, whose conclusions are justified. Overall, I think this is an excellent piece of work.

I have only some minor comments:

1) Title

I still don't see that "after conquest of land" adds anything to the title. Although it is literally true, it is misleading in terms of how long after conquest of land these innovations happened.

2) Abstract & PIN2

The abstract still refers to PIN2, but given the content of the paper it would be better to refer to "...evolution of gravitropism-specific PINs. The PIN apical/shootward subcellular localization is the major evolutionary invention..."

3) The language is generally of a very good standard, but could use a good proof-read and copy-edit. There are lots of extra "the"s, and also quite a lot of missing "the"s in other places.

4) Line 271: Still refers to PtPIN100

5) Line 255: Still refers to CrPIN314

REVIEWERS' COMMENTS:

Reviewer #1 (Remarks to the Author):

This revised version of the manuscript by Yuzhou Zhang and collaborators (NCOMMS-19-04573A: "Evolution of Fast Root Gravitropism in Seed Plants after Conquest of Land") takes into account most of the comments and suggestions made by the reviewers on the previous draft. In particular, the authors provide new data that show a lack of amyloplast sedimentation in lycophyte and fern roots. They take into account the relative root growth rates in their comparative analyses of root gravitropism between species, and provide a more nuanced view on the evolutionary process, acknowledging the possible involvement of other selective forces. They document the conservation of specific phosphorylation-target sites known to contribute to polar localization of the PIN2 and PINH/G proteins in flowering plants and gymnosperms, and include a new paragraph discussing the important contribution of PIN3, PIN4 and PIN7 proteins in fast root gravitropism in flowering plants and gymnosperms (along with their absence in other plant clades displaying evidence of slower root gravitropism). They also provide a better phylogenetic analysis of PIN proteins that is more compatible with previously published trees. Overall, the changes made to this manuscript have substantially enhanced its quality. As pointed out in my review of the first draft, the data presented here are novel and should be of a broad interest to the scientific community. However, I still have a few concerns about this work, which need more attention:

RESPONSE: We thank again for the reviewer 1's nice comments about our manuscript.

1) The analysis of amyloplast sedimentation in root cells of lycophytes and ferns revealed an inability to sediment in these species under treatments that normally promote their sedimentation in flowering plant columella cells. From this observation, the authors conclude: "These results strongly suggest that, unlike in the flowering plant roots, the amyloplasts in the roots of these basal vascular plants did not evolve to serve as the statolith to participate in gravity sensing". I would recommend more caution in the interpretation of the data. Indeed, independently of whether they sediment or not, these amyloplasts still contain starch and, therefore, are likely to be denser than the cytoplasm. Even if they are tightly anchored to the cytoskeleton, they still have the potential of exerting large, gravity dependent forces onto the cytoskeleton, thereby transmitting information related to the gravity vector to the rest of the cell. If this occurs, the mechanism of gravity sensing would probably be somewhat different from the one used by the columella cells of the cap in flowering plants. However, these non-sedimenting amyloplasts would still contribute to gravity sensing (potentially, at least).

RESPONSE: Thanks for this meticulous comment. Now we rewrite this sentence as follows to avoid the over description and make it much more consistent with our observed results.

"These results strongly indicates that, unlike in flowering plant roots, the gravity sensing machinery with the amyloplast sedimentation along the gravity vector did not evolve in roots of these basal vascular plants"
Page 6 Line 126-128.

2) The discussion on the role of PIN3 in root gravitropism focuses on reinforcing the conclusion that "this PIN clade also evolved to facilitate the fast root gravitropism of the seed plants after their divergence from the fern lineage" (lines 321-323). This conclusion seems to completely miss the fact that this clade is missing not only in the non-seed plants, but also in the monocots. This observation should be incorporated in the evolutionary model discussed in this manuscript.

RESPONSE: Thanks for pointing it out! Now we discuss the loss of the PIN3 in the monocot and propose the possible explanation in the Discussion section (Page 14 Line 326-329), to make this observation be incorporated in our evolutionary model.

3) I continue to strongly believe that the apical-basal nomenclature is highly confusing nowadays when applied to plant biology as it has been used to define opposite sides of the cells / structures in roots by different authors. I would argue that a paper that is likely to target the large and diverse readership of a journal of the stature of Nature Communications should attempt to eliminate any terms that might confuse a section of the readership.

RESPONSE: Corrected as previously suggested. Now we use the nomenclature 'shootward-rootward' to replace the original confusing one 'apical-basal' throughout the manuscript.

4) Most of the quantitative data presented in this manuscript lack any statistical analysis of significance. For instance, statistical analyses (and corresponding symbols) should be added to Figure 3 panel m, Figure 4 panel f, Suppl. Figure 2 (panels a and c), Suppl. Figure 6, panel b, Suppl. Figure 9, panel B, and Suppl. Figure 10, panels c and e.

RESPONSE: Corrected and added. Thanks!

5) The title of Suppl. Figure 4 is misleading. Indeed, this figure shows only the location of amyloplasts in root tips, but it does not address the actual site of gravity sensing in these roots. This would require additional functional analyses that have not been carried out in this work.

RESPONSE: Thanks for pointing it out. To avoid the misleading, we now correct the heading of Supplementary Fig. 4 as the "Exclusive root apex-specific amyloplast location in the seed plants." with the original phrase 'gravity perception' replaced by the 'amyloplast location'.

Reviewer #2 (Remarks to the Author):

I previously reviewed this manuscript, and thought there were very exciting data, but that the interpretation of the data needed to be improved. The authors have addressed all my major concerns, and I feel this is now a much more coherent manuscript, whose conclusions are justified. Overall, I think this is an excellent piece of work.

RESPONSE: We thank again for reviewer 2's good comments on our manuscripts.

I have only some minor comments:

1) Title

I still don't see that "after conquest of land" adds anything to the title. Although it is literally true, it is misleading in terms of how long after conquest of land these innovations happened.

RESPONSE: Now the title is revised to the "The evolution of fast root gravitropism in seed plants" as the reviewer previously suggested.

2) Abstract & PIN2

The abstract still refers to PIN2, but given the content of the paper it would be better to refer to "...evolution of gravitropism-specific PINs. The PIN apical/shootward subcellular localization is the major evolutionary invention..."

RESPONSE: Revised as suggested, Page 2 Line 22-23. Thanks very much!

3) The language is generally of a very good standard, but could use a good proof-read and copy-edit. There are lots of extra "the"s, and also quite a lot of missing "the"s in other places.

RESPONSE: We have checked throughout the manuscripts to adequately use the word 'the'.

4) Line 271: Still refers to PtPIN100

RESPONSE: Now Corrected (now Line 274).

5) Line 255: Still refers to CrPIN314

RESPONSE: Now Corrected (now Line 259). Thanks for the careful reading!